# Uncovering the organization of neural circuits with Generalized Phase Locking Analysis

**Shervin Safavi**[1,2], **Theofanis I. Panagiotaropoulos**[1,3], **Vishal Kapoor**[1,4], **Juan F. Ramirez-Villegas**[1,5], **Nikos K. Logothetis**[1,4,6], **Michel Besserve**[1,7] *

1 Department of Physiology of Cognitive Processes, Max Planck Institute for Biological Cybernetics, Tübingen, Germany, 2 IMPRS for Cognitive and Systems Neuroscience, University of Tübingen, Tübingen, Germany, 3 Cognitive Neuroimaging Unit, INSERM, CEA, CNRS, Université Paris-Saclay, NeuroSpin center, 91191 Gif/Yvette, France, 4 International Center for Primate Brain Research (ICPBR), Center for Excellence in Brain Science and Intelligence Technology (CEBSIT), Chinese Academy of Sciences (CAS), Shanghai 201602, China, 5 Institute of Science and Technology Austria (IST Austria), Klosterneuburg, Austria, 6 Centre for Imaging Sciences, Biomedical Imaging Institute, The University of Manchester, Manchester, United Kingdom, 7 Department of Empirical Inference, Max Planck Institute for Intelligent Systems and MPI-ETH Center for Learning Systems, Tübingen, Germany

* michel.besserve@tuebingen.mpg.de

**Data Availability Statement:** The present work is a methodological study, and no new data have been generated for it. We provide the MatLab codes to regenerate all the figures from scratch (except the

## Abstract

Despite the considerable progress of *in vivo* neural recording techniques, inferring the biophysical mechanisms underlying large scale coordination of brain activity from neural data remains challenging. One obstacle is the difficulty to link high dimensional functional connectivity measures to mechanistic models of network activity. We address this issue by investigating spike-field coupling (SFC) measurements, which quantify the synchronization between, on the one hand, the action potentials produced by neurons, and on the other hand mesoscopic "field" signals, reflecting subthreshold activities at possibly multiple recording sites. As the number of recording sites gets large, the amount of pairwise SFC measurements becomes overwhelmingly challenging to interpret. We develop *Generalized Phase Locking Analysis* (GPLA) as an interpretable dimensionality reduction of this multivariate SFC. GPLA describes the dominant coupling between field activity and neural ensembles across space and frequencies. We show that GPLA features are *biophysically interpretable* when used in conjunction with appropriate network models, such that we can identify the influence of underlying circuit properties on these features. We demonstrate the statistical benefits and interpretability of this approach in various computational models and Utah array recordings. The results suggest that GPLA, used jointly with biophysical modeling, can help uncover the contribution of recurrent microcircuits to the spatio-temporal dynamics observed in multi-channel experimental recordings.

## Author summary

Modern neural recording techniques give access to increasingly highly multivariate spike data, together with spatio-temporal activities of local field potentials reflecting integrative processes. We introduce GPLA as a generalized coupling measure between these point-

schematic figures and figures contain raw experimental data) in the public repository https://github.com/shervinsafavi/gpla.git.

**Funding:** All authors were supported by the Max Planck Society. M.B. was supported by the German Federal Ministry of Education and Research (BMBF) through the funding scheme received by the Tübingen AI Center, FKZ: 01IS18039B. N.K.L. and V.K. acknowledge the support from the Shanghai Municipal Science and Technology Major Project (Grant No. 2019SHZDZX02). The funders had no role in study design, data collection and analysis, decision to publish, or preparation of the manuscript.

**Competing interests:** The authors have declared that no competing interests exist.

process and continuous-time activities to help neuroscientists uncover the distributed organization of neural networks. We develop statistical analysis and modeling methodologies for this measure and demonstrate its interpretability in simulated and experimental multi-electrode recordings.

This is a *PLOS Computational Biology* Methods paper.

## Introduction

Understanding brain function requires uncovering the relationships between neural mechanisms at different scales [1–3]: from single neurons to microcircuits [4, 5], from microcircuits to a single brain area [6], and from a single area to the whole brain [7, 8]. Therefore, it is crucial to develop tools to investigate the cooperative phenomena that can connect different levels of organization, such as oscillatory neuronal dynamics [3]. These oscillations are hypothesized to support neural computations [9–13] and various cognitive functions [14–17] and manifest themselves in Local Field Potentials (LFP), a mesoscopic extracellular signal [18] resulting from ionic currents flowing across the cellular membranes surrounding the electrode. LFP oscillatory activity partly reflects a number of subthreshold processes shared by units belonging to underlying neuronal ensembles and responsible for the coordination of their activity [19–22]. As a consequence, a large body of empirical investigations support the functional relevance of LFP oscillations (for reviews, see [19, 20, 22–24]).

The synchronization between spiking activity and LFP has been observed experimentally and led to different theories of cognitive functions. Notably, attention has been hypothesized to rely on interactions between various neural populations coordinated by network oscillations [25–28]. However, the exact network mechanisms that govern these spike-field coupling (SFC) phenomena remain elusive, and notably the role played by the phase of the coupling [29–31]. Arguably two major obstacles are (1) that common SFC measures are pairwise [32–38], which makes the information conveyed about the network increasingly difficult to grasp as the number of pairs of channels increases and (2) the link between SFC measurements and the underlying neural circuit mechanisms is not well understood, a problem that we will call *biophysical interpretability*.

Elaborating on the first obstacle, pairwise analyses are arguably suboptimal for modern neural recording datasets, as state-of-the-art multichannel electrophysiology systems [39–41] allow simultaneous recording of hundreds or even thousands of sites [24, 40, 42, 43]. This represents an unprecedented opportunity to study the large scale collective organization binding spiking activity of individual units with mesoscopic spatio-temporal dynamics (e. g. wave patterns [44]), but at the same time generates high dimensional matrices of pairwise connectivity measurements from which extracting interpretable information is a challenge in itself. Moreover, statistical analysis and significance assessment of parallel spike trains is also challenging (see [45] for a review) and requires novel, more computationally *efficient* approaches in the high dimensional setting.

Elaborating on the second obstacle, this shift in data dimensionality also offers the opportunity to go beyond the phenomenological model of synchronization between these activities, to achieve a precise account of how network properties shape the detailed spatio-temporal

characteristics of collective phenomena. Such account fits the general framework of *constitutive mechanistic explanation* in Science [46, 47], where the occurrence of a phenomenon is explained based on mechanisms governing components of the system under study. However, this requires simple enough links to be established between high-dimensional observations and properties of biophysical models, which remains largely unaddressed for large-scale neural networks [48–51]. To reflect this issue, we will call *biophysical interpretability* the extent to which quantities derived from brain activity measurements can be related to properties of a given biophysical model. While methodologies have been developed that elegantly combine frequency-based analysis, multi-variate methods and dimensionality reduction techniques [52–54], the outcomes typically remain only interpreted as mere phenomenological models of brain activity, broadly built on the idea that the brain processes information through networks of oscillators coupled at different frequencies. As such, these approaches suffer from critical limitations when it comes to providing mechanistic insights that relate to a biologically realistic understanding of the underlying neural circuits. To go beyond this limitation, model reduction approaches are used in physics and biology, including neuroscience [55, 56], to simplify complex models. While these methods have initially mostly been developed to reduce computational complexity, they have also started to be used to foster interpretability [57]. The characteristics of biophysical model reductions and advanced multivariate data analyses need to be carefully chosen to allow biophysical interpretability of high-dimensional measurements, and have not yet been explored for the case of multivariate spike-field coupling.

Thus, we develop a "Generalized Phase Locking Analysis" (GPLA) to address the need for an efficient multivariate method that, in conjunction with suitable neural models, allows biophysical interpretations of spike-field coupling data. GPLA characterizes and assesses statistically the coupling between the spiking activity of large populations of units and large-scale spatio-temporal patterns of LFP.

The benefits of this approach are demonstrated in detail with network models with increasing levels of complexity and biophysical realism, and ultimately with neural data. Each of these settings is designed to demonstrate certain strengths of GPLA. First we provide a theoretical motivation and illustrate how to interpret the outcome of the analysis with toy models. Then we illustrate the statistical benefits of GPLA over uni-variate methods with several simple generative models of spike and LFP. Thereafter, we turn to biophysical interpretability of GPLA using an analytical reduction of two population (excitatory-inhibitory) neural field models. This mechanistic interpretation is exemplified in computational models of hippocampal and cortical neural networks. In particular, we show how studying the phase of GPLA can untangle the contribution of recurrent interactions to the observed spatio-temporal dynamics. Based on these results, application of GPLA to Utah array recordings finally provides evidence of strong feedback inhibition in the macaque prefrontal cortex.

## Results

The overarching motivation of this work is to foster a neuroscientific understanding of experimental data by leveraging biophysical models, i.e. models that comprise equations accounting for the biophysics of neural activity and measurements. Broadly construed, models can be ranked according to the chosen trade-off between realism and complexity, with on one end simplified (e.g. low dimensional, linear, . . .) analytically tractable models, whose dependency on biophysical properties is easily characterized, and on the opposite end, highly detailed computational models, where the role of biophysical parameters can only be assessed by running costly simulations. However, when it comes to the use of models for interpreting data, another key aspect coming into play is the choice of *quantities of interest* (QoI) (following

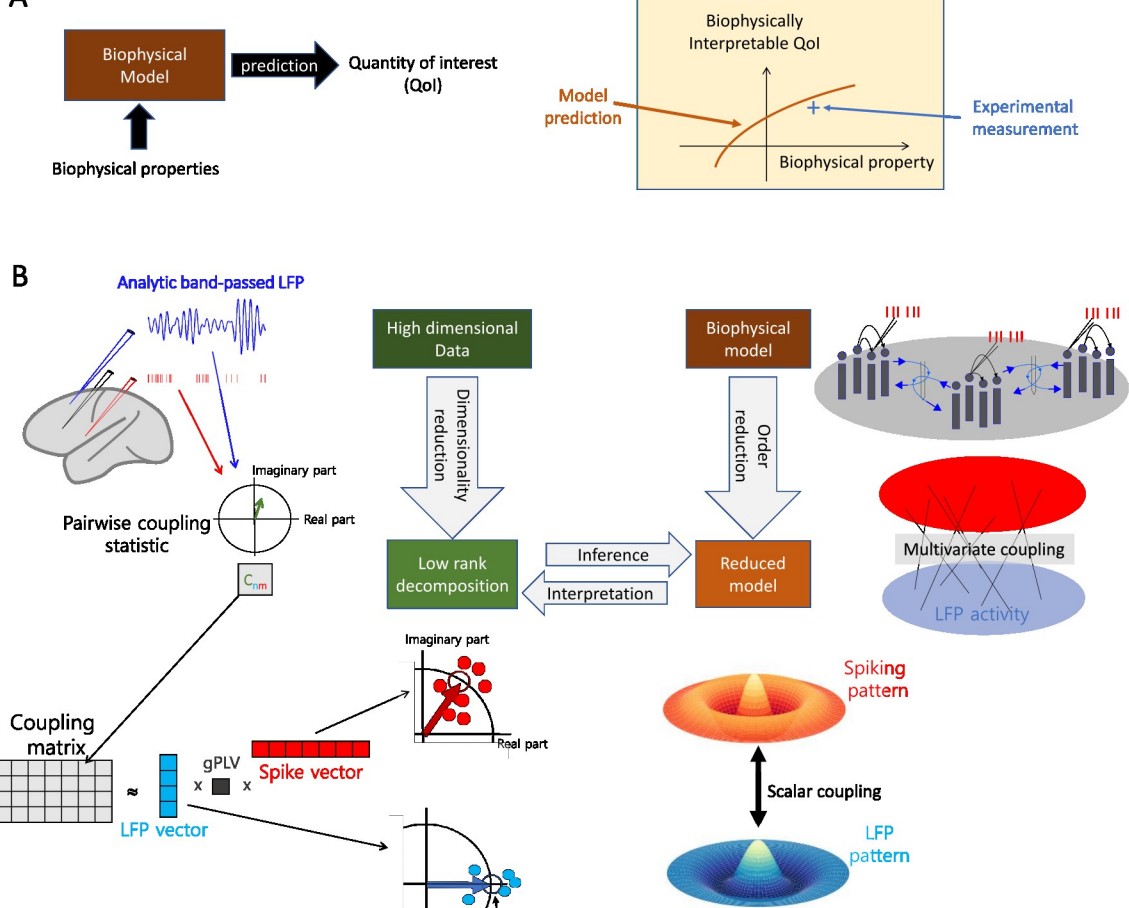

**Fig 1. Interpretability of multivariate SFC through GPLA. (A)** Schematic for the concept of *biophysical interpretability*. A biophysical model allows to make predictions about some observable quantity derived from neural data, that we call *Quantity of Interest* (QoI). The QoI is biophysically interpretable whenever its variations can be explained by changes in some property of the model. **(B) (Top-left)** A coupling matrix is estimated from electrophysiology data by gathering complex SFC estimates of all spike-LFP pairs in a rectangular matrix. Coefficients ($C_{nm}$) contain information similar to complex-valued PLV up to a scaling factor: the magnitude indicates the strength of coupling, and the angle reflects the average timing of the spike occurrence within the period of the corresponding LFP oscillation. **(Bottom-left)** The coupling matrix can be approximated using its largest singular value and the corresponding singular vectors. Singular vectors represent the dominant LFP (blue array) and spiking patterns (red array) and the singular value ($d_1$), called generalized Phase Locking Value (gPLV), characterizes the spike-field coupling strength for the phenomenon under study and the chosen frequency. The magnitude of each vector entry indicates a relative coupling of the corresponding unit/channel, and the phase indicates the relative timing with respect to other units/channels. By convention, the phase of the LFP vector coefficients' average is set to zero, such that the phase of the spike vector average reflects the overall phase shift of the spike pattern with respect to the LFP pattern. **(Top-right)** A biophysical model accounts for the underlying network connectivity and dynamics, as well as the measurement process that leads to the collected data. This leads to a theoretical account of multivariate spike-field coupling. **(Bottom-right)** Model reduction entails simplifying assumption, leading to a low-rank description of the coupling in the model based on the key mechanistic parameters. This description is compared to the left-hand side low-rank decomposition, obtained from experimental data, to infer parameters and interpret the data. All clip art in this figure was designed and drawn by authors M.B. and S.S..

[57]), which are used to assess qualitatively or quantitatively the match between experimental data and candidate models. Indeed, some QoIs may be easier to interpret with a given biophysical model than others, in the sense that they have a more straightforward dependency on biophysical parameters. This leads to us introducing the notion of *biophysical interpretability*, illustrated in Fig 1A and defined as follows: *a QoI is biophysically interpretable for a particular*

*model whenever we can identify biophysical properties of the model that influence the QoI in a simple way* (the simpler, the more interpretable). By "simple", we refer to the low complexity of the functional relationship between the properties and the QoI. To fix ideas, functional relations containing only few biophysical parameters and monotonous functions will be considered simple, although the choice of notion of complexity may be adapted to the case at hand. This interpretability property clearly depends on the model employed, and its benefits for analyzing experimental data relies on the assumption that the chosen model captures well relevant properties of the ground truth mechanism, as it is the case for any attempt at understanding empirical data through mechanistic models. The validity of this assumption can never be fully guarantied and which aspects to incorporate in the model to address a particular question should be assessed based on the literature. In this work, we rely on neural field models [55], which lend themselves to analytical treatment, while still accounting for key aspects of the underlying biophysical network mechanisms. As neurophysiology experiments rely on an increasing number of recording channels, the choice of QoI has to be made from a space of increasingly large dimensions, and we argue that the notion of biophysical interpretability can guide this choice. Specifically, we will consider QoIs that quantify SFC, and a multidimensional generalization of it.

## Generalizing SFC to the multivariate setting

QoIs characterizing the coupling between signals originating from a pair of recording channels are commonly used in Neuroscience. On the one hand, we consider the instantaneous spike rate $\lambda(t)$ of a given unit; and on the other hand, oscillatory activity $L_f(t)$ is derived from the LFP by band-pass filtering in a narrow band of center frequency $f$. We assume $L_f(t)$ is the complex analytic signal representation of this oscillation, computed using the Hilbert transform [58], such that $L_f(t) = a_f(t)e^{i\phi_f(t)}$, where $a_f(t)$ and $\phi_f(t)$ are the instantaneous amplitude and phase of the oscillation, respectively. The coupling between these signals can be characterized by the covariance

$$c(f) = \langle\lambda(t)L_f(t)\rangle = \langle\lambda(t)a_f(t)e^{i\phi_f(t)}\rangle = |c|e^{i\Phi_c} = |c|(\cos(\Phi_c) + i\sin(\Phi_c)), \tag{1}$$

where the $\langle\cdot\rangle$ indicates averaging across time and experimental trials. The complex number $c$ then reflects the strength of coupling through its modulus $|c|$, and the dominant LFP phase of spiking through its argument $\Phi_c$ (see Fig 1(Top-right)). This coupling measure is a modification of the Phase-Locking Value (PLV) [33] (see Eq 13), and differs from the latter mainly through the incorporation of the amplitude of the oscillation in the averaging, and the absence of normalization by the spike rate. We consider the coupling defined in Eq 1 as a base quantity to explain our approach, while normalization will be addressed at the end of this section. Although $\lambda(t)$ is a priori unknown, $c(f)$ is straightforward to estimate based on observed spike times, leading to the empirical estimate denoted $\hat{c}(f)$ (see [59]). However, as more channels are recorded, the number of PLV values to consider increases dramatically, which poses a challenges to the their interpretation. Alternatively, using dimensionality reduction to synthesize the information provided by this large number of couplings may provide a more interpretable picture of the functioning of the underlying circuits.

As illustrated in Fig 1B, Generalized Phase Locking Analysis (GPLA) is introduced as a dimensionality reduction technique to estimate the key properties of the coupling matrix $\mathbf{C}(f)$ consisting of the pairwise couplings between a large number of units and LFP channels at frequency $f$. The estimate $\hat{\mathbf{C}}(f)$ of the coupling matrix based on spiking activity is defined as follows. Given $N_m$ spike times $\{t_k^m\}$ for unit $m$ and the analytic signal $L_f^n(t)$ that is filtered around

frequency $f$ for LFP channel $n$, the $(n, m)$ coordinate of the coupling matrix's estimate $\hat{C}(f)$ is computed by summing the values taken by the analytic signal at all spike times (see Fig 1B (Top-left)),

$$\hat{C}(f)_{n,m} = \sum_k L_f^n(t_k^m).$$

(2)

Next, as schematized in Fig 1B(Bottom-left), the coupling matrix is approximated by the term with the largest singular value $d_1$ of its Singular Value Decomposition (SVD) leading to

$$\hat{C} = UDV^H = \sum_k d_k \boldsymbol{u}_k \boldsymbol{v}_k^H \approx d_1 \boldsymbol{u}_1 \boldsymbol{v}_1^H,$$

(3)

where $\boldsymbol{v}_k^H$ indicates the transpose conjugate of the vector $\boldsymbol{v}_k$. In this expression, the singular value $d_1$ is a positive scalar, that we will call generalized Phase Locking Value (gPLV), and which quantifies the magnitude of the coupling. In order to assess the effect size of $d_1$, but also to perform significance analysis, normalization of the coupling matrix $\hat{C}(f)$ is typically performed, as described in section *GPLA for electrophysiology data* of Materials and methods. In particular, the LFP time series can be whitened beforehand, such that the outcome of GLPA is invariant to the LFP power at each frequency. Table 1 indicates for which experiments such a normalization is applied. The associated complex valued singular vectors in this factorization will be respectively called the *LFP vector*, defined as $\boldsymbol{u} = \boldsymbol{u}_1$ and the *spike vector*, defined as $\boldsymbol{v} = \boldsymbol{v}_1$. As illustrated in Fig 1B (bottom), the spike vector indicates the pattern of coordinated spiking activity most coupled to LFP oscillations, while the LFP vector reflects the dominant spatio-temporal pattern of LFP involved in this coupling. Importantly, based on Eq 3, the difference between the phases of each component of $\boldsymbol{u}$ and $\boldsymbol{v}$ reflects the phase lag between spiking and LFP activities for the respective channels and units. Notably, this implies that all units and all LFP channels with non-vanishing coefficients in spike and LFP vectors have correlated activities at this frequency, as will be further illustrated in the next section. In particular, two units with non-zero coefficients in the spike vector typically have correlated spike rates at this frequency. Multiplication of both singular vectors by the same unit complex number leads to the exact same approximation as Eq 3, reflecting that GPLA only measures the relative phase between LFP and spikes. To resolve this ambiguity in our analyses, we adopt the convention of setting the phase of the average across all components of the LFP vector $\langle \boldsymbol{u} \rangle = \frac{1}{n_c} \sum_k u_k$ to zero, as illustrated in Fig 1B (bottom). As a consequence, the phase of the mean of the spike vector coefficients $\langle \boldsymbol{v} \rangle = \frac{1}{n_u} \sum_k v_k$ reflects the difference of mean phases between spiking and LFP activities. See section *GPLA for electrophysiology data* in Materials and methods for more details.

**Table 1. Summary of normalization by spike count and whitening application in all figures.**

| Figure num. | Spike count normalization type | Whitening applied | Equations |
|---|---|---|---|
| Fig 2 | $1/N$ | No | 16 |
| Fig 3 | $1/N$ | No | 16 |
| Fig 4 | $1/\sqrt{N}$ | Yes | 17 |
| Fig 6 | $1/\sqrt{N}$ | Yes | 17 |
| Fig 7 | $1/\sqrt{N}$ | Yes | 17 |
| Fig 8 | $1/\sqrt{N}$ | Yes | 17 |

Importantly, we demonstrate GPLA can simultaneously be applied to a neural field model (analytically or in simulations), to yield a reduced biophysical model (see Fig 1B(Right)). The outcome of GPLA applied to neural data can then be interpreted based on this reduction, as explained in section *Reduction of complex models based on linear response theory*. Notably, this can be exploited to study the key network characteristics giving rise to the observed spike-field coupling. Here, we demonstrate the possibility of this hybrid approach merging modeling and analysis for a certain class of generative models, while further development is needed to extend it to a more general setting.

## Illustration of GPLA and statistical benefits over univariate SFC

We first illustrate how GPLA provides an intuitive phenomenological model of the coupling between the population of spiking units and LFPs. We use three simulations in which a transient global LFP oscillation recorded in a single channel (Fig 2A) modulates the firing probability in 18 spike trains (attributed to neuron-like units). As described on the left column of Fig 2C–2F, models instantiate (1) a global oscillation driving a synchronous population of neurons (2) wave-like discharges of neurons (similar to the case of "delayed excitation from a single oscillator" described by [44]) (3) groups of cells that fire together predominantly at three distinct phase values of the LFP. For comparison, a fourth simulation is performed with no coupling. Exemplary spike trains for each model are displayed in the second column of Fig 2C–2F overlaid on the magnified version of the LFP oscillation.

For all models, the coupling is well reflected by the gPLV magnitude obtained from these simulations, as shown in Fig 2B. Moreover, the phase of the spike vector components resulting from GPLA summarizes the coupling structure in an intuitive way in Fig 2C–2F (right column), showing: (1) all components collapse to a single phase, (2) evenly distributed phases of the spike vector coefficients over a 180 degrees interval, (3) three distinct phases, (4) an isotropic phase distribution, as predicted by mathematical analysis [59].

These simple simulations demonstrate how to interpret the spike vector. Because there is a single LFP channel in this setting, GPLA straightforwardly combines univariate coupling measures of each unit. However, statistical analysis of GPLA is different from the univariate case, as we show next with a setting similar to the above model (3) of Fig 2E but with weaker coupling of individual neurons to the oscillation, leading to values at the edge of significance (assessed with the surrogate-based test, see section *Significance assessment of gPLV* in Materials and methods). An illustrative simulation in the case of low noise and large number of observed spikes is shown in Fig 3B, together with the corresponding spike vector in Fig 3C, providing results similar to Fig 2E.

For quantitative analysis, we consider the setting of a single LFP channel and a handful of neurons are the focus of the analysis (Fig 3A–3E). Such recordings are still common and valuable in human electrophysiology experiments for understanding cognition [60, 61]. While pooling the spikes from all units into a single spike train to get a *pooled Phase-Locking-Value* (pPLV) may result in a higher statistical power, it requires the distribution of the locking phase to be homogeneous across units (e.g., in the case of Fig 2C, but not for Fig 2D and 2E). In contrast, GPLA exploits the spike times from multiple neurons to assess the global coupling between spikes and LFPs without requiring such homogeneity. We ran 5000 simulations with only 3 units and compared the coupling assessment based on PLV, pPLV, and gPLV. Fig 3D represents the estimated PLVs, with averages matching the couplings obtained with a larger number of spikes in Fig 3C. Performance of each measure is assessed based on its detection rate, which is defined as the percentage of simulations for which significant coupling is detected, as assessed using spike-jittered surrogate data (see Materials and methods section

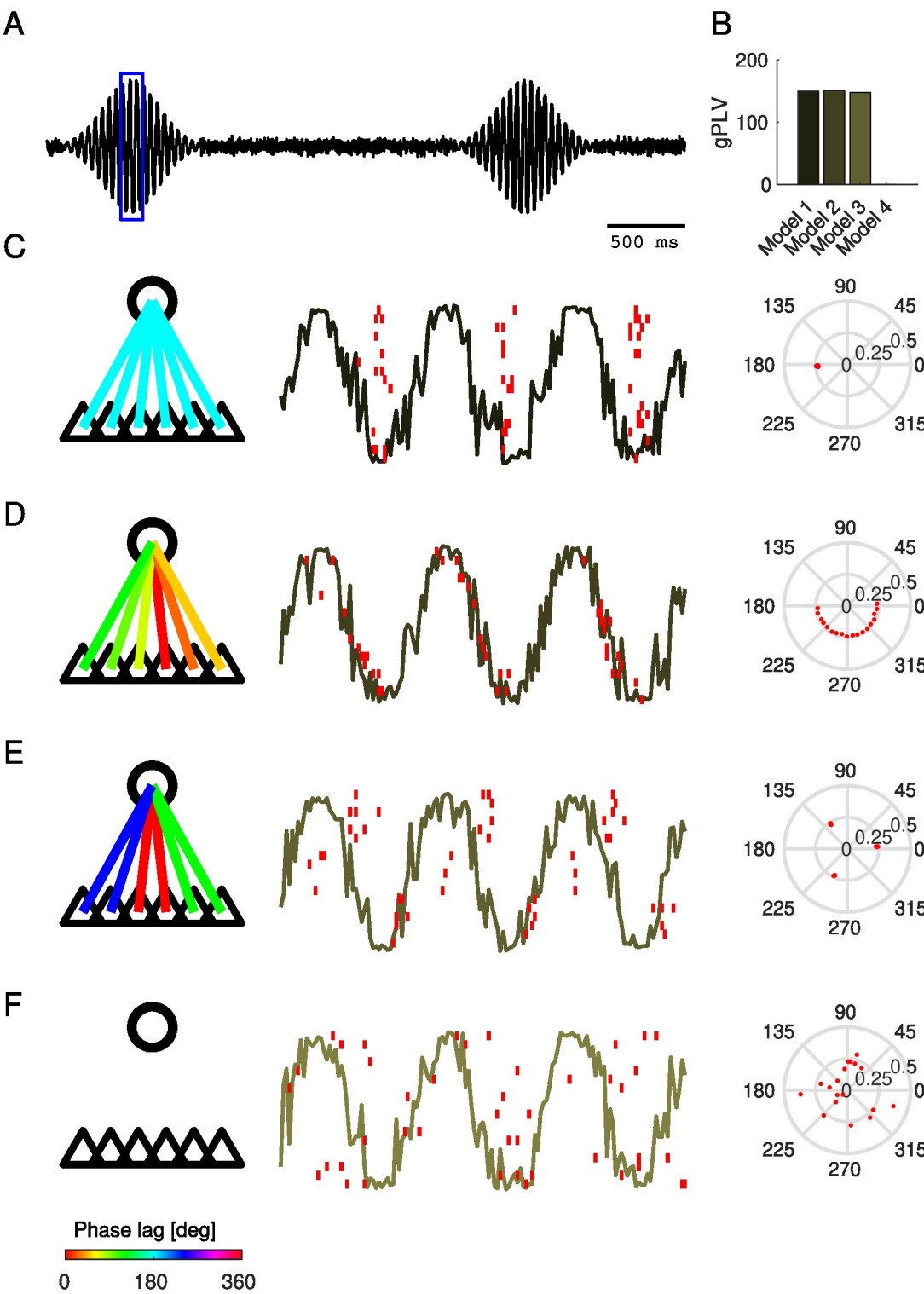

**Fig 2. Illustration of GPLA on simple simulations.** **(A)** Normalized amplitude of LFP-like oscillatory signals. **(B)** gPLVs for different models demonstrated in C-F **(C-F)** Various scenarios of spike-LFP coupling. Left: schematic representation of the modulating LFP oscillation (circle), and 6 representative neuron-like-units (indicated by the triangles). The color of each connecting line indicates the locking phase (see bottom colorbar for color code). Center: LFP-like signals within the window specified by the blue box in A and spikes are represented by overlaid red vertical lines. Right: resulting spike vector is represented in the third column. **(C)** Spiking activity globally synchronized to the trough of the LFP oscillation. **(D)** Sequential discharge of

spikes coupled to the LFP. **(E)** Three clusters of neurons discharge at different phases of the LFP oscillation (a similar model was also used in Fig 3). **(F)** Spiking activity uncoupled to LFP oscillation (independent homogeneous spike trains). Also see Table A in S1 Appendix for a methodological summary.

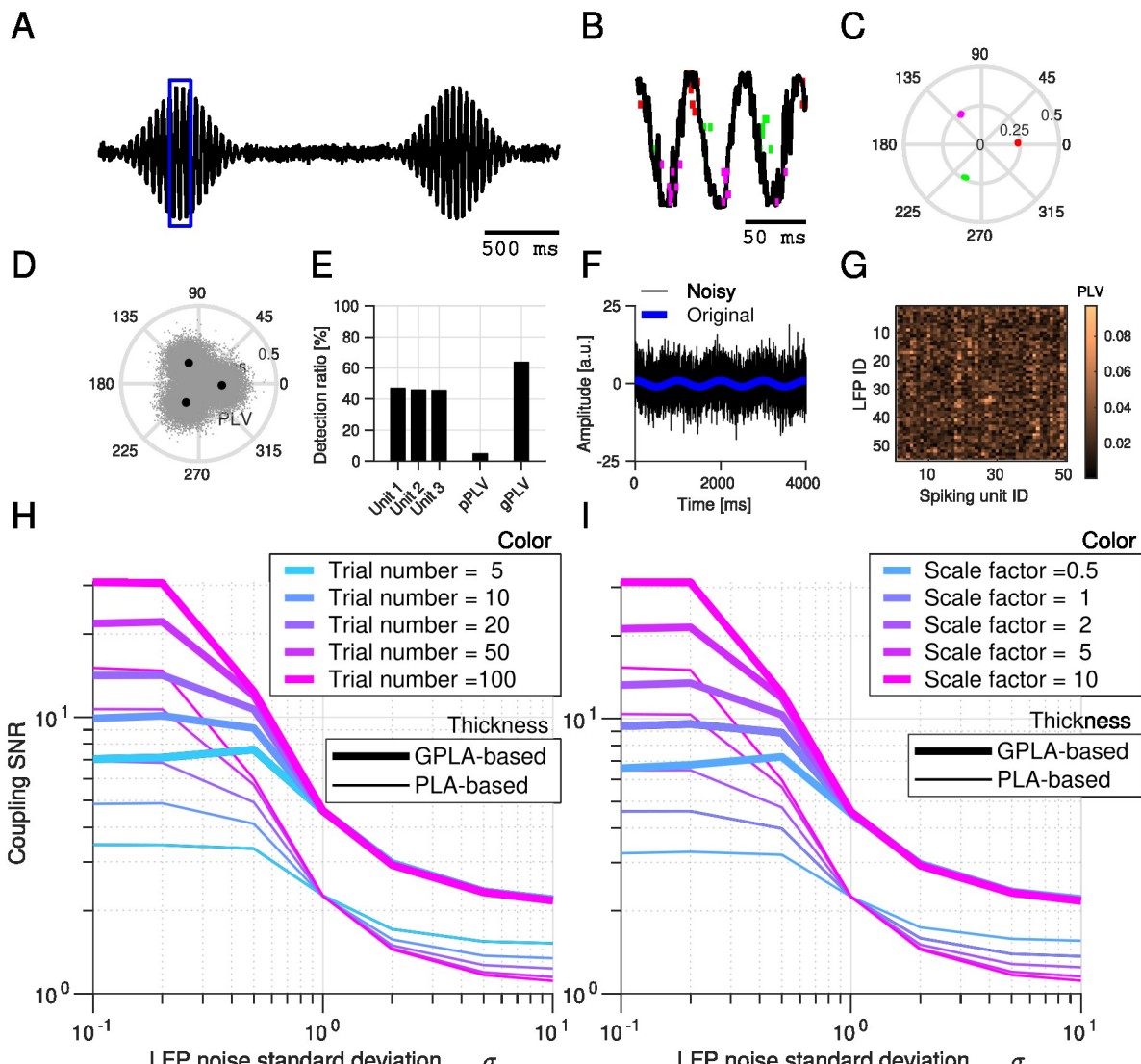

**Fig 3. Comparison of GPLA and uni-variate spike-field coupling. (A)** Normalized amplitude of LFP-like transient oscillatory signal with additive Gaussian white noise (used in the first simulation). **(B)** LFP-like signal and overlaid spike raster (colored vertical lines—colors indicate each population of units with common locking phase) within the window specified by the blue box in (A). **(C)** Spike vector coefficients in the complex plane (colors correspond to B). Each dot represents one coefficient of the spike vector corresponding to a single neuron (note that within each cluster, dots are overlapping as they are similarly coupled). **(D)** Complex PLVs represented in the complex plane. Angles indicate the locking phase and the radius of the PLV. The gray point clouds indicate the PLV of multiple simulations and larger black dots indicate the average values. **(E)** Performance comparison (in percentage of simulations with significant coupling) of PLV, pooled PLV (pPLV) and gPLV, for three individual neurons. **(F)** Example oscillation, original (blue trace) and noisy (black trace) used in the second simulation. **(G)** Example coupling matrix related to simulation with a large amount of noise ($\sigma = 5$) **(H-I)** Comparison of GPLA-based and PLA-based estimation of PLVs for (H) different number of trials and (I) different levels of firing rate. Signal-to-Noise Ratio (SNR) is defined as the ratio of coupling strength (PLV) to estimation error (the difference between estimated PLV and the ground truth). Also see Table A in S1 Appendix for a methodological summary.

*Significance assessment of gPLV*) and with a significance threshold of 5%. As it is demonstrated in Fig 3E, gPLV detection outperforms the competing approaches (PLV and pPLV).

Beyond improved detection of a significant overall coupling, GPLA-based estimation of pairwise couplings based on the approximation of Eq 3 may also be more accurate than individual estimates when the data is very noisy and multivariate, benefiting from the SVD procedure to disentangle noise from the ground truth coupling (see Eq 16 for the expression of normalized coupling matrix used here). To demonstrate this, we performed another simulation (Fig 3F–3I), similar to the above, but using 50 LFP channels containing oscillations driving spike-LFP coupling, contaminated by different levels of noise (i. e. adding Gaussian noise with different variances to the transient oscillation, see section *Simulation of phase-locked spike trains* in S1 Appendix for details), and modulating the firing rates of the units, lower firing rates leading to a larger amount of estimation variance for the PLV [59]. An example LFP trace with (black) and without (blue) noise is exemplified in Fig 3F and an example coupling matrix in the presence of noise is also illustrated in Fig 3G. In this case, the ground-truth coupling matrix has rank one, as all the units are locked to a single frequency (coupling matrices with higher ranks can also be achieved and analyzed in a similar way, see Fig 4). We ran the simulations with different amounts of LFP noise (indicated on the x-axis of Fig 3H and 3I), computed the coupling coefficients (similar to Fig 3G) and compared it to ground truth (based on Equation S2 in S1 Appendix). The Signal-to-Noise Ratio (SNR) was defined as the ratio of coupling strength (PLV) to estimation error (the difference between estimated PLV and the ground truth—for more details see section *Computing Signal-to-Noise Ratio* in S1 Appendix) and was used to compare the quality of GPLA-based and univariate estimation (indicated in the y-axis of Fig 3H and 3I). As this simulation demonstrates, the estimation error of the coupling coefficients is larger for the univariate estimation than for the GPLA-based approach for a broad range of noise levels (Fig 3H and 3I). Additionally, we can observe a sharp drop of the estimation quality of the GPLA-based approach as the noise increases, likely reflecting a phase transition phenomenon in high-dimensional random matrices reported in [59]: above some noise level threshold, singular value and vector information cannot be retrieved from noisy observations, while they can be recovered with very good accuracy above it. This property is further exploited in the next section.

## Random matrix theory based fast significance assessment

While in the previous section, GPLA's significance was assessed using surrogate data, this approach is computationally expensive and provides limited insights into the statistical properties of GPLA estimates. We also investigated this question using mathematical analysis, and exploited it to assess more efficiently the significance of multivariate coupling. Singular values and vectors estimated by GPLA have an intrinsic variability due to the stochasticity of spiking activity, which can be investigated through stochastic integration and random matrix theory [62, 63]. In the absence of coupling between spikes and LFP, appropriate preprocessing allows deriving analytically the asymptotic distribution of univariate and multivariate coupling measures [59], including the convergence of the squared singular values to the classical Marchenko-Pastur (MP) law [64]. Based on the MP law, we can define an upper bound on the largest singular values of the coupling matrix that depends only on its dimensions, such that exceeding this bound indicates the significance of the coupling (for more details see Materials and methods section *Analytical test* and [59]), leading to a fast *analytical test*.

We assessed the performance of this test on simulated spikes and LFPs with or without coupling as follows. Briefly, we synthesized multivariate LFP activity by linearly superimposing several oscillations (denoted $O_k(t)$ in Fig 4A) with different multiplicative weights applied for

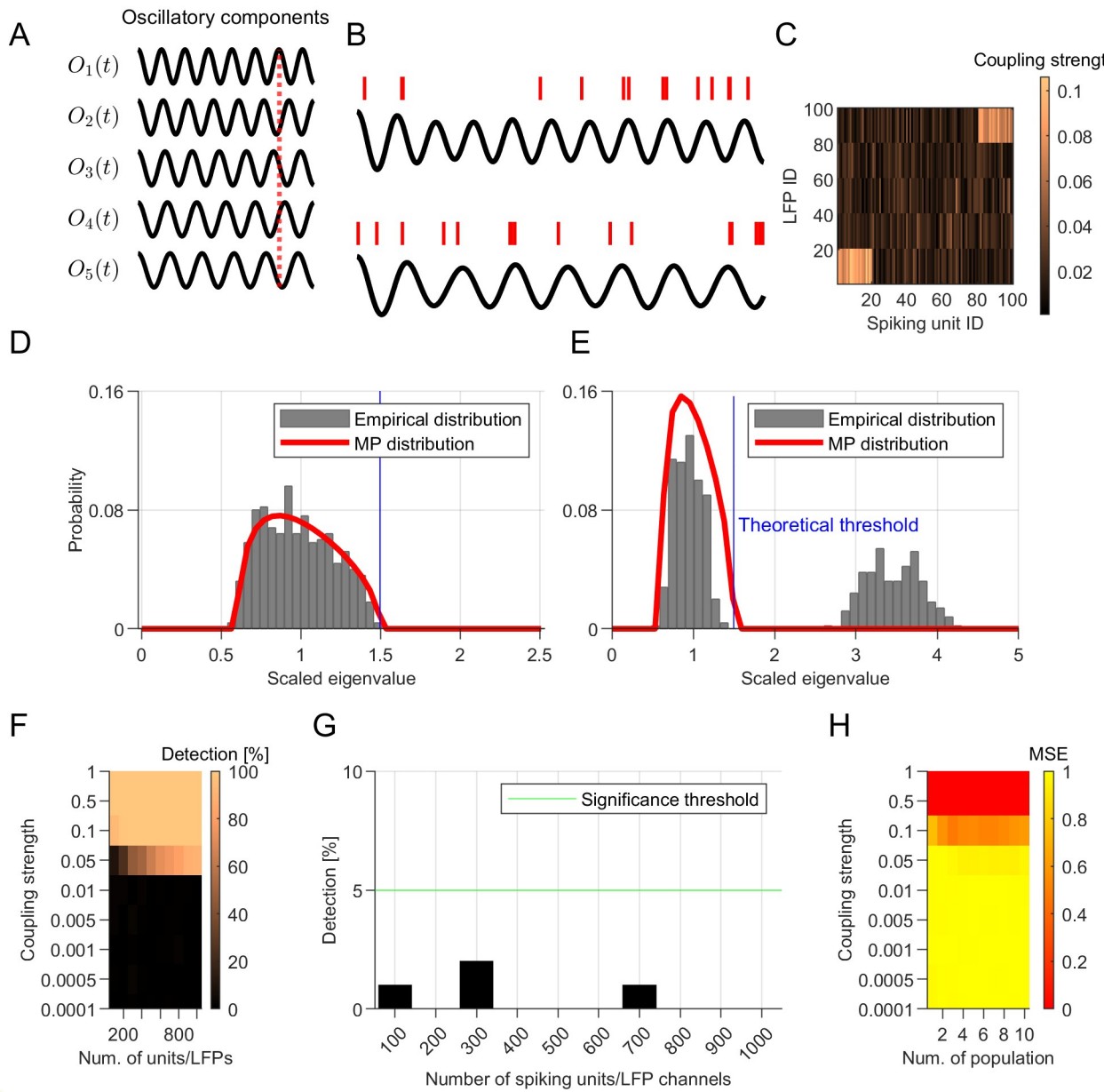

**Fig 4. Statistical analysis of GPLA with a theoretical significance test.** (A) LFPs are synthesized by mixing several oscillatory components ($O_k(t)$). The vertical red line evidences the phase shift between them. (B) Two exemplary spike trains (each from one of the coupled populations) and the corresponding LFPs. In the LFP trace on the top, the oscillatory component with the highest frequency is dominant while the bottom one is dominated by the lowest frequency component. (C) An exemplary coupling matrix for a simulation with two coupled populations. (D-E) Theoretical Marchenko-Pastur distribution (red lines) and empirical distribution (gray bars) for (D) simulation without coupling and (E) with coupling between multivariate spikes and LFP (F) Performance of GPLA for the detection of coupling between spike trains and LFPs for different strength of coupling (y-axis) and different number of spiking units/LFP channels. (G) Type I error for different numbers of spiking units/LFP channels (x-axis), quantified as the percentage of simulations wherein a significant coupling between spike trains and LFPs is detected in absence of ground truth coupling. The horizontal green line indicates the %5 threshold. (H) Mean-squared-error of GPLA-based estimation of the number of populations coupled to LFP for varying coupling strengths (y-axis) and numbers of coupled populations (x-axis). See also Table A for a methodological summary.

each LFP channel and generated the spike trains of each unit with Poisson statistics. As for the coupling between spikes and LFPs, 2/5th of the units were coupled to the LFP oscillations (exemplified in Fig 4B), while the remaining units had homogeneous Poisson spike trains (for details see S1 Appendix, section *Simulation of phase-locked spike trains*). The estimated

coupling matrix computed based on Eq 17 for a simulation with 100 spike trains and 100 LFPs is exemplified in Fig 4C, where we have two coupled populations, one coupled to the lowest-frequency and one coupled to the highest-frequency oscillatory component of the LFP (reflected by the top-right and bottom-left bright blocks of the coupling matrix in Fig 4C and sample spike trains and LFP in Fig 4B).

By computing the SVD of the coupling matrix after application of the preprocessing explained in S1 Appendix, section *LFP pre-processing*, we can obtain a spectral distribution for the squared singular values, which matches the prediction of the theory (Fig 4D and 4E). In the absence of coupling between spikes and LFP signals (Fig 4D), the distribution of the eigen-values closely follows the MP law and in the presence of coupling, the largest eigenvalue exceeds the significance bound predicted by random matrix theory (RMT) (see section *Analytical test* in Materials and methods for more details).

We further quantified the type I and II error of this analytical test. For type II error, we ran the simulations with non-zero coupling between spikes and LFP signals. As shown in Fig 4F, GPLA was able to detect a significant coupling between spike and LFP even when the coupling strength was as small as 0.05 (no coupling corresponds to 0 strength and perfect coupling cor-responds to 1). These results also show the performance of the test does not degrade with the increasing dimension of the data through the number of recording channels (Fig 4F). This is in contrast with assessing individually the significance of pairwise couplings, for which correc-tion for multiple comparisons (e.g., Bonferroni) would typically lead to a degradation of the power of the test as the number of units/LFPs increases. This is particularly relevant for weaker couplings, as they may lose significance after correction for multiple comparisons. Additional-ly, we quantified the type I error of the test by running simulations with no coupling between spikes and LFP and quantified the number of false positives. Our results show that our analyti-cal test has a small ($< 5\%$) false positive rate (Fig 4G).

We also quantified the performance of the method for estimating the number of popula-tions coupled to different rhythms. In this simulation, the number of coupled populations can be determined by the number of significant singular values (see the section *Simulation of phase-locked spike trains* in S1 Appendix and section *Analytical test* in Materials and methods for more details). Similar to the simulation explained earlier (Fig 4A–4C), we simulated multi-ple (1–10) non-overlapping cell assemblies synchronized to different LFP rhythms (with dif-ferent frequencies within a narrow range of 11–15.5 Hz). When the coupling was larger than a minimum strength of 0.5, the method was able to capture the number of populations with very low error, $MSE < 0.015$ (Fig 4H).

## Neural field modeling of SFC

While the above results have addressed GPLA's outcome from a statistical perspective, its bio-physical interpretation requires modeling the underlying neural network dynamics. The basis for this interpretation will be a two-population *neural field model*: a spatially distributed rate model of the activity of two interacting homogeneous populations: excitatory pyramidal cells (*E* population) and inhibitory interneurons (*I* population) [65, 66]. The model is governed by three basic input-output relations (see S1 Appendix, section *Analytical neural field modeling of spike-field coupling*) and depicted in Fig 5A: (1) the dynamics of the average somatic mem-brane potentials $V_E$ and $V_I$ of each population is governed by exogenous post-synaptic cur-rents $\eta$ originating from other cortical or subcortical structures as well as recurrent excitatory and inhibitory post-synaptic currents (EPSC and IPSC) $s_E$ and $s_I$; (2) the population spike rates $\lambda_E$ and $\lambda_I$ are a function of their respective membrane potentials; and (3) EPSC and IPSC are each controlled by the spike rate of their afferent population (E and I respectively). In the

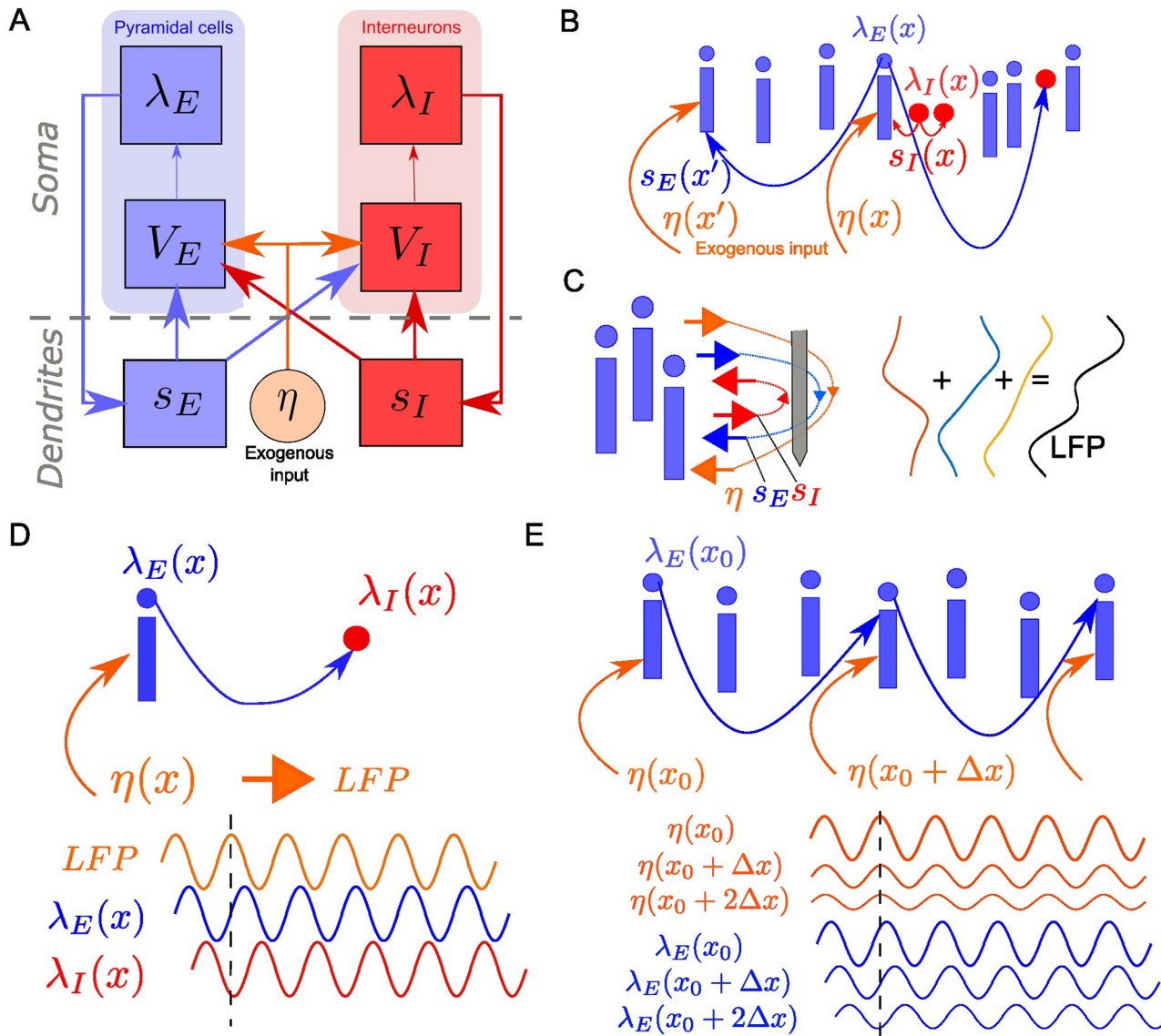

**Fig 5. Generative model of spike-LFP coupling. (A)** A two-population neural field model of neural dynamics. $V_k$, $\lambda_k$ and $s_k$ indicate respectively somatic membrane potential, firing rate and post-synaptic current for Excitatory (k = E) and Inhibitory (k = I) populations. $\eta$ indicates the exogenous input to the circuit. Arrows indicate the causal dependence between variables of the model. **(B)** Schematic representation of the model's connectivity: local inhibition and long range excitation, together with the driving by exogenous synaptic currents. **(C)** Schematic representation of the contribution of postsynaptic currents to the electric field, affected by the spatial distribution of synapses over the dendritic tree and the geometry of pyramidal cells. From left to right: Schematic representation of pyramidal neurons, electric field, electrode (gray bar), contribution of each current (EPSC, IPSC and exogenous current, leak current is also contributing to LFP but is not shown) to the LFP profile along the electrode's axis **(D)** Simple microcircuit structure leading to a temporal ordering of the local activities of different kinds $LFP \rightarrow excitation \rightarrow inhibition$ **(E)** Simple microcircuit structure leading to a temporal ordering of activities of the same kind across space: the location receiving stronger exogenous input leads other locations, such that amplitude gradient leads to phase gradients.

context of large-scale recordings, the neural population can be distributed across one or several spatial directions. Following a classical approximation depicted in Fig 5B, inhibitory connections are assumed local [67–69], such that coupling between cells surrounding distinct recording locations happens exclusively through excitatory axons ($s_E(x)$ may depend on $\lambda_E$ at other spatial locations than $x$) as well as through common exogenous input current $\eta$.

For the LFP $L(t)$, resulting from the conduction of trans-membrane currents in the extra-cellular space, we assume the contribution of currents flowing through the membrane of inter-neurons is negligible, based on the weakness of the anisotropy induced by their dendritic geometry across the population [70–72]. The LFP thus results exclusively from pyramidal cell's membrane currents. Which currents (IPSC, EPSC, leak current, exogenous current) affect the most the recorded LFP at a given spatial location depends on multiple factors: the geometry of the cells, the distribution of synapses (inhibitory, excitatory, exogenous) onto them, and the geometry of the electrodes [19, 20, 23]. Fig 5C provides a schematic of how the differentiated location of synaptic boutons over the dendritic tree may result in variable algebraic contributions of each type of current to each recording channel.

In the following simplistic but biophysically interpretable connectivity scenarios, this model provides insights on how the underlying microcircuit parameters influence SFC properties. Fig 5D depicts a microcircuit receiving exogenous inputs exclusively onto the pyramidal cells' dendrites (no feedforward inhibition), while *I* cells receive local excitatory inputs, but do not synapse back onto *E* cells (no feedback inhibition). If we assume additionally that sub-threshold activity is dominated by the exogenous input currents and proportional to the measured LFP, then the lag induced by the membrane potential dynamics then results in a positive (frequency-dependent) lag of excitatory activity with respect to the LFP (reflecting the input), while inhibitory activity is itself delayed with respect to excitation. For an exogenous input oscillating at an arbitrary frequency, this implies a phase lag configuration between the (oscillatory) responses of these variables.

Circuit assumptions may also provide insights on how the same variable varies across spatial locations, which we illustrate by extending spatially the previous microcircuit scenario (with no feedforward and feedback inhibition), by adding horizontal E-E connectivity with a decreasing strength as a function of distance (see Fig 5E). If we assume the activity results from a spatially inhomogeneous oscillatory input, with larger input amplitude at a given side (on the left in Fig 5E), the delay induced by membrane dynamics entails the propagation of the activities from one side of the circuit to the opposite. This leads to an interesting relationship between the phase and amplitude of oscillatory activity: the location of the largest amplitude is ahead of time with respect to the neighboring locations with smaller amplitudes. Interestingly, these propagation-like patterns are induced by the assumed network horizontal connectivity, while the input to the structure does not have phase lags at different locations [44]. These simple connectivity scenarios indicate that phase and amplitude of oscillatory activities, which GPLA captures through the spike and LFP vectors, are informative about the underlying microcircuit structure and dynamics. More realistic scenarios must take into account recurrent interactions between cell populations, as we will see in the next sections.

Note that up to this point, the developed neural field models can be used to interpret uni-variate as well as multivariate SFC. When the number of pairs for which SFC can be computed becomes large, a difficulty of a different nature appears: how can we synthesize the interpretations that we get from all these pairs? While ad hoc approaches for selecting relevant pairs to derive interpretations from is an option, we can try instead to establish interpretability of QoIs derived from GPLA, as we found support for its relevance for describing coupling properties of the system as a whole.

## Reduction of complex models based on linear response theory

In order to analyze more complex circuits, a systematic and quantitative way to link model parameters to the coupling between network activities at a given frequency is required. We assume small-amplitude perturbations in the neighborhood of an operating point, such that

the static sigmoidal conversion of membrane potentials into spike rates can be linearized (see section *Analytical neural field modeling of spike-field coupling* in S1 Appendix and [55, 73–75]). This leads to a linear time-invariant model, whose behavior is fully characterized by its amplitude and phase response to oscillatory inputs at each frequency. When considering the coupling between firing rate ($\lambda_E(x_1, t)$) and field ($L(x_2, t)$) at two locations $x_1$ and $x_2$, linearity and time invariance entails the existence of transfer functions (denoted $H_{\lambda_E}$ and $H_L$ respectively), linking the spatial distributions of the time domain Fourier transforms of network activities, denoted $\hat{\lambda}_E(x, f)$ and $\hat{L}(x, f)$, to the one of the exogenous input $\hat{\eta}(x, f)$, as follows:

$$\hat{\lambda}_E(x, f) = \int H_{\lambda_E}(x, x', f)\hat{\eta}(x', f)dx' \quad \text{and} \quad \hat{L}(x, f) = \int H_L(x, x', f)\hat{\eta}(x', f)dx'. \quad (4)$$

Next, this model can be simplified by assuming an approximate space-frequency separability:

$$\hat{\eta}(x, f) \approx n(x)\hat{\epsilon}(f), \text{ at each location } x \text{ and each frequency } f. \quad (5)$$

Using the above transfer functions, this leads to both spike rate and LFP being proportional to the exogenous input, with respective multiplicative coefficients $\psi_E(x, f)$ and $\psi_L(x, f)$ defined as follows:

$$\hat{\lambda}_E(x, f) \approx \int H_{\lambda_E}(x, x', f)n(x')\hat{\epsilon}(f)dx' = \psi_E(x, f)\hat{\epsilon}(f) \quad \text{and} \quad \hat{L}(x, f) \approx \psi_L(x, f)\hat{\epsilon}(f). \quad (6)$$

As a consequence, the coupling between LFP and $E$ spikes at respective locations $x_1$ and $x_2$ writes (up to a multiplicative constant, see section *Details for the low rank approximation of Equation 7* in S1 Appendix)

$$C_{x_1, x_2}(f) \approx \langle \hat{\lambda}_E(x_2, f)^* \hat{L}(x_1, f) \rangle \approx \psi_L(x_1, f)\psi_E(x_2, f)^* \langle |\hat{\epsilon}(f)|^2 \rangle, \quad (7)$$

where $z^*$ denotes the complex conjugate of $z$. This shows that the coupling between $L$ at $x_1$ and $\lambda_E$ at $x_2$ is separable in the spatial variables $(x_1, x_2)$, and characterized by two functions of space: one for the field, $\psi_L$, and one for the excitatory spiking, $\psi_E$. In particular, as $\langle |\hat{\epsilon}(f)|^2 \rangle$ is a positive number, the phase of $C_{x_1, x_2}$ reflects a property of the underlying circuit irrespective of its input, and given by the phase difference between $\psi_L$ and $\psi_E$ at the considered frequency and locations. Importantly, the functions $\psi_L$ and $\psi_E$ also describe the coupling between the same variables at different locations, e.g. $\langle \hat{\lambda}_E(x_2, f)^* \hat{\lambda}_E(x_1, f) \rangle \approx \psi_E(x_1, f)\psi_E(x_2, f)^* \langle |\hat{\epsilon}(f)|^2 \rangle$, such that their phase distribution across locations informs about the spatial functional connectivity of the network. Likewise, $\psi_I$ can be defined for inhibitory activity and merged with $\psi_E$ to describe the rates of all units of both populations.

In practice, $C_{x_1, x_2}(f)$ can be measured at only a finite number of locations, corresponding to electrode channels where $L$, $\lambda_E$ and $\lambda_I$ are recorded. This leads to a rectangular matrix $C(f)$ estimated by multiple pairwise SFC estimations, combining excitatory and inhibitory units. The above separability Eq 5 then implies that $C(f)$ is a rank-one matrix, such that it can be decomposed exactly according to GPLA, where the LFP vector reflects $\psi_L$ while the spike vector concatenating E and I units reflects $\psi_E$ and $\psi_I$. Overall, Eq 6 imply that the spatial distribution of the phase and magnitude spike and LFP vectors is influenced by the underlying network interactions (shaping the transfer functions such as $H_{\lambda_E}$ and $H_L$), as well as by the type of currents that dominate the LFP. As we will illustrate in the next sections, the analysis of these GPLA features across frequencies is thus a rich source of information to validate assumptions about local network organization based on experimental multivariate data.

## Application to spike-field dynamics during sharp wave-ripples

The phenomenon of hippocampal Sharp Wave-Ripples (SWR) is one of the most striking examples of neural activity entangling spike and LFP dynamics in multiple frequency bands, attributable to specific mechanisms of the underlying microcircuit [76]. Specifically, SWRs are brief episodes observed in hippocampal LFP traces combining a high-frequency oscillation (the ripple) to a low-frequency biphasic deflection (the sharp-wave). Moreover, these LFP activities are well known to be synchronized with spiking activity, with each cell-type firing at a specific phase of the ripple oscillation [76], but also with further spike-field couplings at lower frequencies [77].

We use simulations of in-vivo SWR described in [78] in order to demonstrate what insights GPLA can provide about the underlying hippocampal network mechanisms. The model generates realistic spiking and LFP activity in subfields CA1 and CA3, based on populations of two-compartment Hodgkin-Huxley neurons distributed along two distant one dimensional grids representing the strata of each subfield. In this model, the connectivity of CA3 is characterized by strong recurrent excitatory auto-associational $E - E$ connections, together with $E \rightarrow I$ connections and short-range $I \rightarrow E$ and $I - I$ synapses (see Fig 6A for a schematic representation). In contrast, local $E - E$ connections are absent in CA1, but both E and I cells receive feedforward excitation from CA3. LFPs were generated from the total trans-membrane currents using line current density approximation, and measured by two laminar multi-shank electrodes (see S1 Appendix, section *Simulation of hippocampal sharp wave-ripples* for more details).

We first apply GPLA to a single hippocampal subfield, CA1, as various studies suggest SWRs emerge from it in response to afferent CA2- and CA3-ensemble synchronous discharges [79, 80]. In this simulation, LFP and unit recordings are distributed along two orthogonal spatial directions (laminar for LFPs and horizontal for units). We use a total of 157 peri-ripple traces of simulated LFPs and spikes of both populations (inhibitory and excitatory) of duration approximately 1 sec. Exemplary traces of simulated LFP and population firing rate of the CA1 population (pyramidal cells and inhibitory interneurons belonging to CA1) are shown in Fig 6B.

GPLA results for representative frequency bands are provided in Fig 6C–6E and for all bands covering the 1–180Hz interval in S2 Fig. The overall coupling magnitude (gPLV) was significant for all frequencies (Fig 6C), according to both surrogate (based on spike jittering, $p < 0.05$) and analytical (based on random matrix theory) tests. In particular, the strongest coupling was detected in the ripple band (80–180 Hz), in line with results obtained with classical univariate techniques on experimental data [76].

The LFP vectors strongly overlap across frequency bands, and exhibit a biphasic electric potential profile typical of laminar recordings (Fig 6D). This corresponds to the field generated by the dipolar geometric arrangement of sources and sinks in the parallel two-compartment models of pyramidal neurons used for this simulation. To check the quantitative agreement between the LFP vector and the original model of LFP generation in this simulation, we computed analytically the total LFP generated passively by all pyramidal cells using the original LFP simulation code of [78], and assuming all cells have identical trans-membrane currents flowing through their somatic and dendritic compartments (see S1 Appendix, section *Simulation of hippocampal sharp wave-ripples*). While the dendritic current reflects the post-synaptic input of the cell, somatic currents are taken opposite to preserve the charge neutrality of each cell. The resulting theoretical LFP profile of the pyramidal populations are highly similar to the LFP vector (cosine similarity > 0.97 for LFP vector of all three frequencies in Fig 6D). Note that the sign of the LFP vectors' coefficients results from our convention of setting the phase of

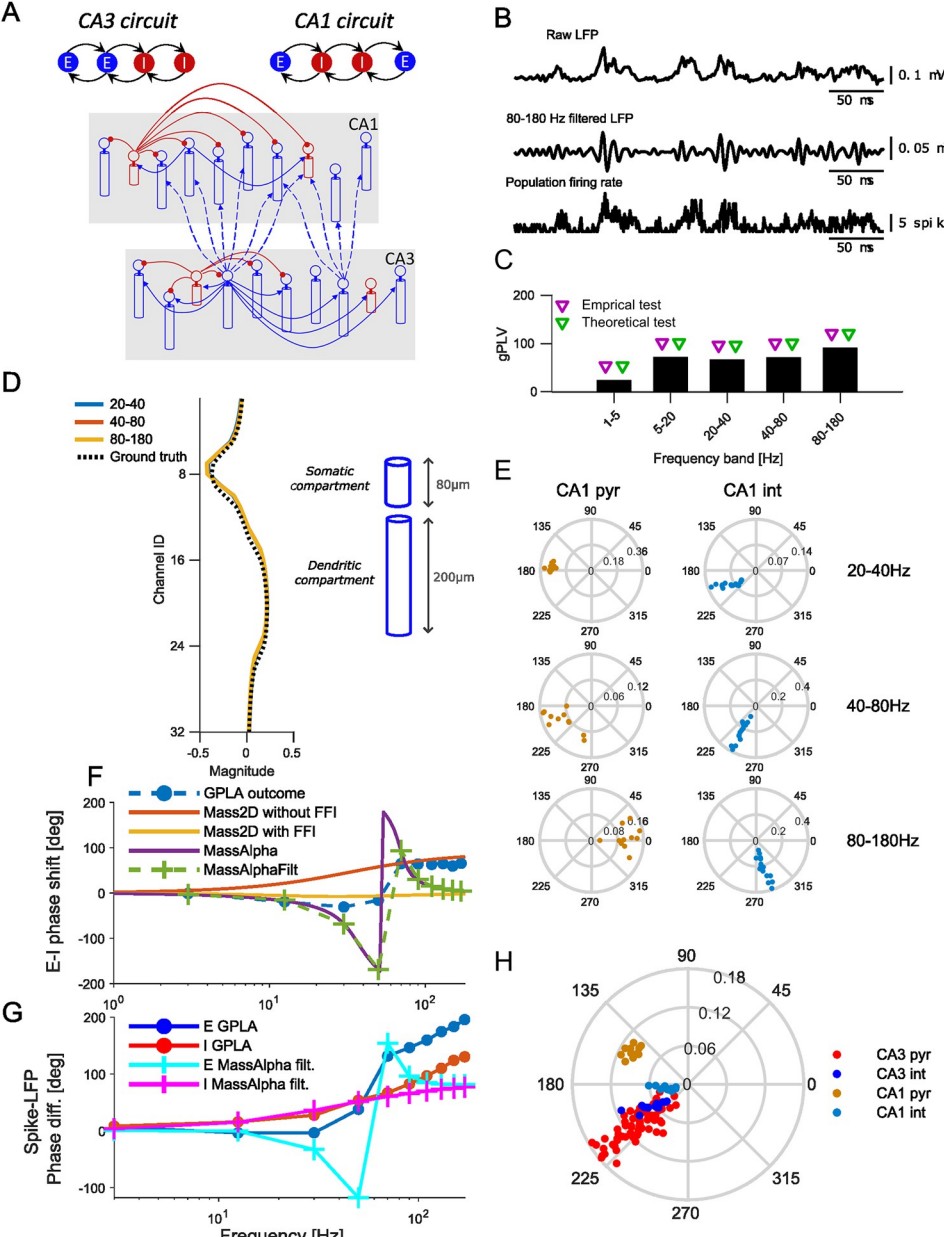

**Fig 6. GPLA of hippocampal SWRs generated by a biophysical model of [78]. (A)** Hippocampal multi-compartment model. Top: Canonical circuits of CA1 and CA3. Bottom: Schematic of the whole model (blue, excitatory connections; red, inhibitory. **(B)** From top to bottom: Example broad band CA1 LFP trace, band-pass filtered trace of the CA1 LFP in ripple band (80–180 Hz), and population firing rate of CA1 neurons. **(C)** CA1 gPLVs. Triangles indicate the significance assessed based on empirical (blue triangles, p<0.05) and theoretical (red triangles) tests. **(D)** LFP vectors for GPLA of CA1 (blue and red curves are overlapping), superimposed to ground truth dipolar LFP profile passively generated by the two compartment models of the pyramidal cell population. The right-hand side schematic illustrates the vertical dimensions of one cell's compartments. **(E)** Spike vector coefficients for CA1 in several frequency bands (left: pyramidal cells, right: interneurons). **(F)** Average phase lag between LFP and spike vectors across frequencies for: outcome GPLA on hippocampal SWRs, theoretical analysis of *Mass2D* (without and with feedforward inhibition) and *MassAlpha* neural mass models. Dashed green line indicate *MassAlpha* filtered over the frequency bands used for GPLA. **(G)** Difference between phases of E and I populations based on GPLA the *MassAlpha* neural mass model filtered in the same bands (IPSP was used as LFP proxy). **(H)** Spike vector resulting from GPLA jointly applied to CA1 and CA3 in the gamma band (20–40 Hz). Related Supplementary Figures: S1 Fig, Use of EPSP as LFP proxy; S2 Fig, Joint GPLA of CA3 and CA1 activities; S3 Fig, Joint GPLA of CA3 and CA1 activities.

its mean to zero (see Fig 1C). As the LFP vector coefficients are divided into two groups of opposite sign, a positive sign is attributed to the set of coefficients that weight the most in the overall sum. In the context of laminar recordings, one could as well adopt a different convention ascribing a fixed sign to coefficients located in the peri-somatic layer (named *stratum pyramidale* in CA1). This would then lead to a sign consistent with classical analyses, e. g., triggered averaging based on spikes or oscillatory peak [81]. These results, inline with recent studies [82], overall suggest that the LFP vector can be exploited to further study the current sources and sinks causing the LFP, e. g. through current source density analysis [83, 84]. Notably, the result of a similar analysis based on uni-variate phase locking analysis leads to a profile which is incompatible with the ground truth (see S4 Fig).

Moreover, the spike vector components' distribution in the complex plane (Fig 6E) supports that both E and I cells are synchronized to CA1 LFP in the ripple band (80–180 Hz), but at different phases, in line with experimental and simulation results [76, 78]. This extends the observation made for one-directional $E \rightarrow I$ coupling in Fig 5D to a more realistic case of recurrent $E - I$ interactions. Interestingly, pyramidal cells can be clearly differentiated from interneurons based only on their components' respective phase in the spike vector, showing that interneurons lead pyramidal cells in lower frequency bands, while drastically switch to the converse in high frequencies (see also Fig 6F). This direct outcome of GPLA avoids the task of choosing a reference LFP channel on an ad hoc basis to compare the phase of univariate couplings of each units relative to it. Moreover, it can be used not only for inferring cell types from experimental data [85], but also, based on its biophysical interpretability, to address mechanistic questions, as we illustrate next.

We focus on the classical question of oscillogenesis, aiming at uncovering the circuit mechanisms responsible for the emergence of fast oscillations. The way it is addressed in the literature is paradigmatic of mechanistic questions: scientists resort to experimentation and modelling to chose between a restricted number of candidate hypotheses. Two classical candidate mechanisms are the Interneuron Network Gamma (ING) relying on the coupling between inhibitory interneurons under tonic excitation [86], and Pyramidal Interneuron Network Gamma (PING) relying on the interaction between excitatory principal cells and inhibitory interneurons [87, 88]. We take advantage of these biophysically realistic hippocampal simulations, for which the ING mechanism has been shown to be the generator of high frequency activity [78], to assess how biophysical interpretability can help decide which of the two above hypothesis is the right one. To do that, we will derive analytically SFC's phase in linearized neural mass models of the microcircuit activity with different levels of complexity: the simplest accounting for PING, and the more complex also accounting for ING.

In line with [74], we first designed the *Mass2D* model, taking into account somatic time constants (resulting from membrane capacitance and leak currents), but neglecting synaptic dynamics (see S1 Appendix, section *Analysis and simulation of two population neural mass models*). As a result, *Mass2D* is a 2 dimensional dynamical system, allowing only PING resonance through the interactions between pyramidal cells and interneurons. As shown in Fig 6F for typical parameters, and demonstrated analytically (see S1 Appendix, section *Analysis and simulation of two population neural mass models*) the predicted phase shift across frequencies could neither account for the driving by interneurons in CA1, nor for phase changes in high frequencies ($> 30 Hz$). Notably, incorporating strong feedforward inhibition (FFI) did not improve the qualitative match between the analytical predictions and GPLA's outcome. The inappropriateness of *Mass2D* is in line with the current understanding of SWR emergence in CA1 through the pacing of pyramidal activity by delayed $I - I$ interactions [89], as *Mass2D* does not account for them.

The emergence of oscillations through $I - I$ interactions is well understood mathematically, showing that sufficiently strong delayed recurrent inhibition gives rise to resonance or sustained oscillations [90]. We account for this ING mechanism in an extension of the *Mass2D* model, the *MassAlpha* model, by including an additional synaptic delay and/or a synaptic time constant for $I - I$ synapses [74], through the use of so-called *alpha synapses*. (see S1 Appendix, section *Analysis and simulation of two population neural mass models* for details). Interestingly, the resulting sign of the phase shift between $E$ and $I$ populations of this model is now in qualitative agreement with GPLA estimation (Fig 6F), exhibiting a reversal in the lead-lag relation between populations as frequency grows, thereby providing more support for the ING oscillogenesis hypothesis than for PING, in line with evidence provided in the original study [78]. The SFC phase is thus biologically interpretable for the chosen family of neural mass models, is the sense that a phase reversal across the frequency axis appears when lagged I-I interactions responsible for ING are introduced. Because this phase reversal also appears in simulations exhibiting ING that rely on a much more complex model (Hodgkin-Huxley neurons instead of neural masses), this supports the idea that biophysical interpretations of SFC based on our simplified models may generalize to more realistic settings and to experimental recordings.

Another interesting property of the network is the phase shift between each individual population and the LFP, which is simply reflected in the phases of the spike vector coefficients averaged across each population ($E$ and $I$), due to our chosen phase convention (see Eq 23). Given that the LFP is a linear combination of all post-synaptic currents of the network, we can leverage biological interpretability of GPLA to evaluate which of these currents is the most representative of the observed spike-LFP phase relation. As shown in Fig 6G, the choice of the IPSP as an LFP proxy in the *MassAlpha* model accounts qualitatively, as frequency increases, for (1) monotonous phase increase of the I population, (2) the phase slope reversal of the E population (see S1 Fig for a comparison with using the EPSP as LFP proxy). In contrast, using EPSP as an LFP proxy still fails to reproduce these two aspects (see S1 Fig), illustrating how GPLA, beyond microcircuit dynamics, may also help address the cellular underpinnings of experimentally observed LFP [91]. This overall suggests that GPLA combined with neural mass modeling of a structure can provide insights into the microcircuit dynamics underlying phenomena as complex as sharp-wave ripples, despite neglecting many biophysical details. We however emphasize that we restricted ourselves to a qualitative comparison of GPLA features for choosing from a restricted set of biophysical models, which best matches the ground truth mechanisms. This approach holds potential for designing a full-fledged GPLA-based model selection tool, whose development is left to future work.

Importantly, GPLA can also provide further insights when concurrent recordings from multiple regions are available. It allows investigating the coordination of spiking activities across structures without relying on an arbitrary choice of reference LFP channel (also see the analysis of neural data for a realistic demonstration, S8 Fig), by automatically extracting a multi-channel LFP activity (reflected by the LFP vector) that relates the most to spiking activities at a given frequency. We illustrate this by running GPLA jointly on spikes and LFPs from both CA1 and its afferent structure CA3, using the exact same model as above. Fig 6H depicts coefficients of the resulting spike vector, showing CA1 and CA3 neurons are all coupled to the field activity with cell-type-specific phases in the gamma band (20–40 Hz) (see S3 Fig) that are consistent with the GPLA obtained from individual structures (see S2 Fig). This notably suggests that the gamma activity has a dominant coherent component spanning the two structures consistently with current hypotheses that this rhythm supports communication between subfields during memory trace replay [78, 92].

### Application to spatio-temporal patterns of neural field models

One context where biophysically interpretable multivariate methods such as GPLA hold potential is the analysis of cortical spatio-temporal dynamics. Horizontal connectivity is believed to endow many regions with distributed information processing capabilities [30, 44, 93]. However, how underlying connectivity properties relate to experimentally observed multi-channel recordings remains largely elusive. We assessed the ability of GPLA to address this question by first simulating electrode array recordings of a piece of cortical surface with a 2D neural field model, as described in Fig 5. We used an exponentially decaying horizontal excitatory connectivity with a spatial scale constant $r_0 = 440\mu m$, following recent analyses of cortical recordings [91]. The spatio-temporal dynamics were down-sampled spatially on a grid with a step size $\Delta x = 800\mu m$, representing the inter-electrode distance of a putative electrode array of 1.2$cm$ size (see S1 Appendix, section *Analysis and simulations of neural field models* for details). The field is stimulated by a synchronous excitatory exogenous input with a narrow (1.4 mm STD) isotropic Gaussian spatial amplitude distribution reaching its maximum at the center of the field. We compared the spatio-temporal dynamics for two choices of connectivity for which the input-free network has a stable equilibrium. First, we consider the *weak inhibition* case (Fig 7A), for which inhibitory (I) cells have weak feedback inhibition ($I \rightarrow E$), relative to the self-excitation caused by $E - E$ horizontal connections. The resulting activity is akin to stochastic fluctuations, due to the exogenous input, around a *stable node* equilibrium. Second, in the *strong inhibition* case (Fig 7B), the larger excitability of inhibitory neurons strengthens their influence on excitation and leads to activity fluctuating around a *stable spiral equilibrium*, reflecting a tendency of perturbations to oscillate around this point (Fig 7B) [94]. In both cases, the computed excitatory population rate is used to simulate the spike train of one excitatory unit per spatial electrode on this grid, in line with the observation that excitatory units are more easily detected experimentally due to their open field configuration [95]. GPLA is then computed between this excitatory spiking activity and different LFP proxies. The results in Fig 7C–7H are computed using the total EPSP resulting from horizontal E-E connections as LFP proxy (i. e. excluding exogenous excitation). We observe key differences between the GPLA of the two systems, predicted by linear response theory (see S1 Appendix section *Analysis and simulations of neural field models*).

First, as reflected in the gPLV values (Fig 7C), spike-field coupling appears stronger in the lower frequency bands in the case of weak recurrent inhibition, while in the case of strong recurrent inhibition we observe a stronger coupling at intermediate frequencies. Notably, the peak of spike-field coupling in intermediate frequencies for strong inhibition is in line with models of the prefrontal cortex with the same enhanced feedback inhibition [96], exhibiting a resonance in the beta range (25Hz).

Second, as demonstrated in the previous neural mass model simulation, the global spike-LFP phase shift may also be informative about the underlying neural circuits. We can compute the average phase shift between the spike and LFP vectors as a function of the frequency band to see a clear difference between the two models. Strong recurrent inhibition leads to phase advance of the spiking activity in the low frequency, in contrast with the weak recurrent inhibition case showing a consistent lag of excitatory spiking across frequencies (Fig 7E).

Third, the relationship between the spatial variations of modulus and phase of the spike vector is different across these two networks. In the simulation with strong recurrent inhibition, the phase of spike vector coefficients as a function of their modulus for the frequency band associated with maximum gPLV for each model indicates that the phase of the spike vector coefficients decreases (i. e. the oscillation lags further relative to the LFP) for larger modulus ($p < 10^{-4}$, F-test on the linear regression model; $N = 69$), whereas, in the simulation with weak

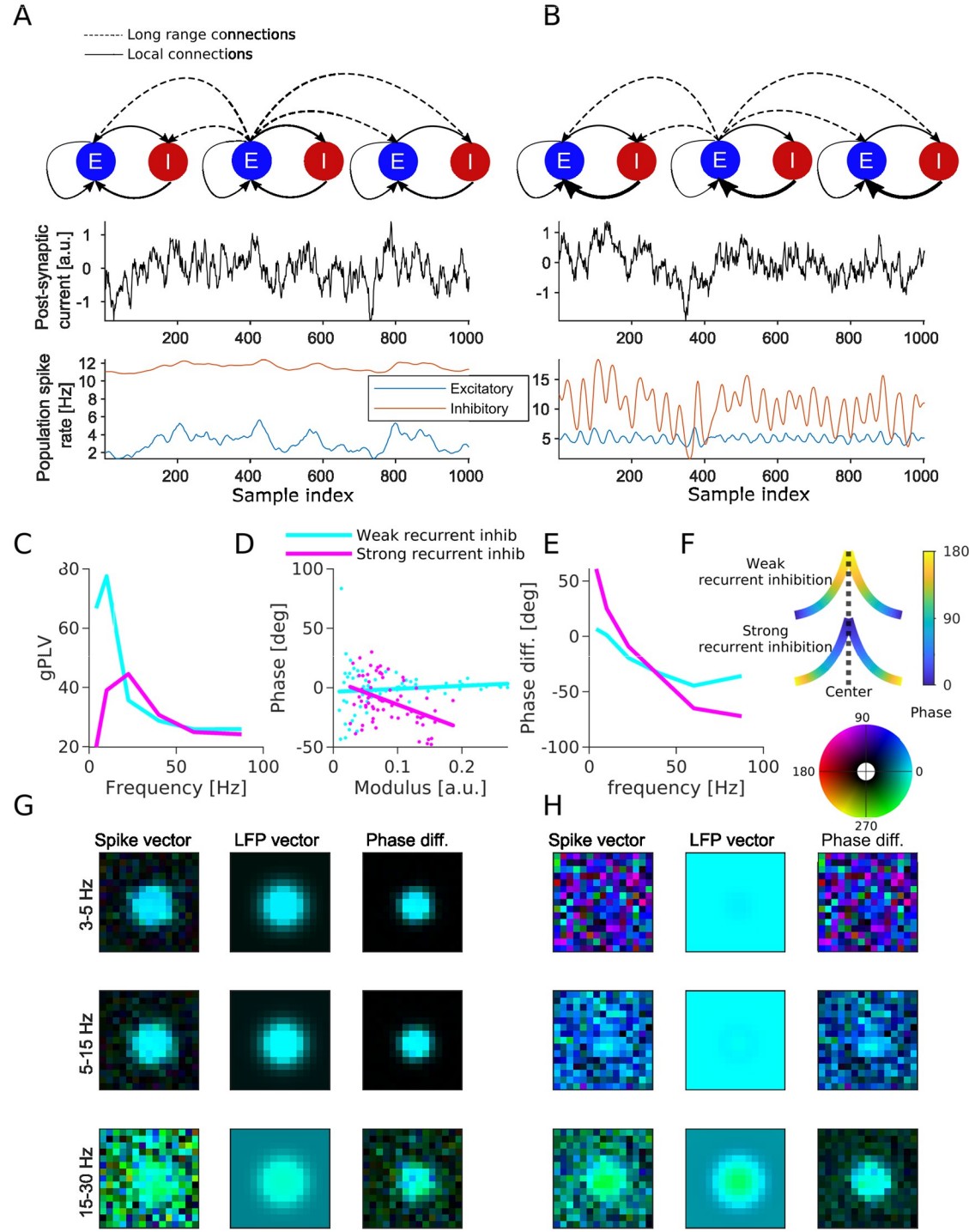

**Fig 7. Neural field simulation using EPSP as LFP proxy. (A)** Simulation with weak recurrent inhibition. Example time course at center location for exogenous input (top), E- and I- populations rates (bottom). **(B)** Same as A for strong recurrent inhibition. **(C)** gPLV as a function of frequency for both models. **(D)** Phase of spike vector coefficients as a function of their modulus for the frequency band yielding maximum gPLV for both models (each dot one coefficient, and the continuous lines are plotted based on linear regression). **(E)** Shift between averaged phase of spike vector and averaged phase of LFP vector, as a function of frequency. **(F)** Schematic of the spike vector's phase gradient in the two models according to Eq 8. X-axis is the distance from center and y-axis is the connectivity strength. Line color indicates the phase according to the colorbar on the right. **(G)** Resulting GPLA in 3 frequency bands (indicated on the left) for weak recurrent inhibition (model schematized in A). **(H)** Same as G for strong recurrent inhibition (model schematized in B). In both G and H, color of pixel code the values of spike/LFP vector coefficients, with colorbar on top of H.

Colors are represented in HSV mode, in which a complex number ($re^{i\phi}$) is represented by hue and brightness of a pixel. Hue of a pixel indicates the phase ($\phi$) and the brightness of a pixel indicates the magnitude ($r$). Related supplementary Figures: S5 Fig, Phase-modulus relation dependency on level of inhibition; S6 Fig, GPLA using IPSP as LFP proxy.

recurrent inhibition, phase is not correlated with modulus ($p > 0.3$, F-test on the linear regression model; $N = 69$) (Fig 7D).

This last difference between the two connectivity cases can be directly interpreted based on the spatial maps of spike vector coefficients across the array. Indeed, models exhibit a different radial phase map in both situations (Fig 7G and 7H), reflecting how phase changes as magnitude decreases when going away from the center (the location with the largest input). This gradient can be predicted by theoretical analysis of a one dimensional neural field, as we show in detail in S1 Appendix, section *Spatio-temporal phase analysis in 1D*. Briefly, the spike vector can be approximated by the spatial convolution of the input spatial pattern at a given temporal frequency *f* by a kernel of the form

$$k(x) = e^{-|x|a(f)} = e^{-|x|\text{Re}[a(f)]}e^{-i|x|\text{Im}[a(f)]} .\tag{8}$$

The first term of this kernel has a negative real number multiplied by distance in the exponential that makes the activity decay away from the locations where exogenous input is the highest, as intuitively expected from the horizontal connectivity of the circuit. For the second term of the product in Eq 8, the imaginary number in the argument of the exponential enforces a spatial phase gradient in response to the input, which depends on the sign of the imaginary part of *a*. If this sign is positive, responses at the location of the highest input will be ahead of time with respect to their surrounding in the considered band, as reflected by their larger spike vector phase in the top illustration of Fig 7F. On the contrary, if Im[*a*] is negative, locations with the highest input are lagging behind (bottom illustration of Fig 7F). Interestingly, these spatial features of the spike vector can be related to the biophysical parameters of the neural field model. Indeed, we can show that the frequency-dependent complex number *a* (*f*) that controls this behavior satisfies the approximate relation (valid at low frequencies, see S1 Appendix section *Spatio-temporal phase analysis in 1D*

$$a^2 \approx \frac{1}{r_0^2}\left[1 + v_{E\to I}v_{I\to E} - v_{E\to E} - i2\pi\tau f\left(2v_{E\to I}v_{I\to E} - v_{E\to E}\right)\right].\tag{9}$$

with $r_0$ the above defined spatial scale of excitatory horizontal connectivity, $v_{P\leftarrow Q}$ the magnitude of synaptic connectivity from population $Q$ to $P$. It can be deduced from this expression that the sign of the imaginary part of *a* (same as for $a^2$) will depend on the relative strength of recurrent inhibition onto pyramidal cells, controlled by $v_{E\leftarrow I}v_{I\leftarrow E}$, with respect to recurrent excitation controlled by $v_{E\leftarrow E}$. Intuitively, having no recurrent inhibition leads to Im[*a*] $> 0$, and classical propagation, mediated by excitatory horizontal connections, away from the location that received an input. In contrast, large recurrent inhibition leads to $2v_{E\leftarrow I}v_{I\leftarrow E} >> v_{E\leftarrow E}$ and Im[*a*] $< 0$. This can be interpreted as a tendency of recurrent inhibition to "suppress" the input that created the response, generating a "wave" converging back to the points where the input was highest. The theory also predicts that large values of $v_{E\leftarrow I}v_{I\leftarrow E}$, as used in the strongly recurrent simulation, can generate strong phase gradients. In contrast, linear stability constrains the values of $v_{E\leftarrow E}$ to remain small, reflecting our choice for the simulations, and resulting in a comparatively moderate slope for the weakly recurrent case. More quantitatively, we further analyzed in S5 Fig the relation between the complex number *a* resulting from a linear approximation of our simulated neural field models, and the linear regression coefficient of the phase-modulus analysis performed in (Fig 7D), for four choices of recurrent inhibition parameters (see Table 2), ranging

**Table 2. List of 2D neural field model parameters.**

| Parameter name | Symbol | Value *(for each level of recurrent inhibition)* | | | |
|---|---|---|---|---|---|
| | | *weak* | *lower med.* | *upper med.* | *strong* |
| E membrane time constant | $\tau_E$ | $20ms$ | $20ms$ | $20ms$ | $20ms$ |
| I membrane time constant | $\tau_I$ | $20ms$ | $20ms$ | $20ms$ | $20ms$ |
| E-E synaptic strength | $\tilde{v}_{E \to E}$ | 0.2 | 0.2 | 0.2 | 0.2 |
| I-I synaptic strength | $\tilde{v}_{I \to I}$ | 0 | 0 | 0 | 0 |
| $E \to I$ synaptic strength | $\tilde{v}_{E \to I}$ | 0.2 | 0.2 | 0.2 | 0.2 |
| $I \to E$ synaptic strength | $\tilde{v}_{I \to E}$ | 1 | 1 | 1 | 1 |
| E excitability | $\chi_E$ | 1 | 1 | 1 | 1 |
| I excitability | $\chi_I$ | 0.1 | .33 | 1 | 3.33 |
| E sigmoid threshold | $V_{th,E}$ | 0 | 0 | 0 | 0 |
| I sigmoid threshold | $V_{th,I}$ | 0 | 1 | 5 | 5 |
| E maximum rate | $Q_E$ | $20Hz$ | $20Hz$ | $20Hz$ | $20Hz$ |
| I maximum rate | $Q_I$ | $20Hz$ | $20Hz$ | $20Hz$ | $20Hz$ |

from weak to strong inhibition. The result exhibits a clear monotonous relation between the regression coefficient and Im[$a$] as well as $\frac{\text{Im}[a]}{\text{Re}[a]}$. Note however that this relation is not one to one, as $a$ characterizes the properties of a kernel that is convolved to the exogenous input to the structure to yield the spike vector, thereby resulting in a spatial smoothing of the phase.

Overall, contrasting multiple cases shows that modifications of the strength of feedback inhibition are reflected not only in the dominant frequency of spike-LFP synchronization (Fig 7C), but also in the spike-LFP shifts of the GPLA results (Fig 7E), and in the relationship between modulus and phase of spike vector coefficients (Fig 7D). Notably, these observations are being made in the absence of specific oscillatory activity nor spatial phase gradient of the exogenous input (which influences the activity synchronously across the array). Therefore, it supports that the observation of complex coordinated activity, such traveling waves-like phase gradients, may emerge from local recurrent interactions in the recorded regions, instead of resulting from the passive driving by spatio-temporally coordinated activity originating from other brain regions.

As it has been argued in the literature that LFP activity may in some cases reflect inhibitory activity [91], we also provide GPLA results when taking the IPSP activity as LFP proxy in S6 Fig. The variations of GPLA features across the frequency axis witness clear differences with respect to the results of Fig 7C–7E, in particular when it comes to the phase difference between spike and LFP phases. This suggests that GPLA also provides information that allow to infer which neural processes are reflected in LFP activity.

## Analysis of Utah array data in the prefrontal cortex

The biophysical interpretability of GPLA features demonstrated in the context of neural field simulations suggests it can provide mechanistic insights about experimental recordings of spatio-temporal cortical activity. Indeed, electrode arrays are able to record the activity of hundreds of units and LFP channels spatially distributed along the cortical surface, and GPLA can be used to link these activities to recurrent cortical circuits, believed to play a key role in information processing. We apply GPLA to Utah array ($10 \times 10$ electrodes, inter-electrode distance $400\mu m$) recordings from the ventrolateral prefrontal cortex of one anaesthetized rhesus monkey (see Fig 8A). LFP signals were preprocessed as described in S1 Appendix, section *Animal*

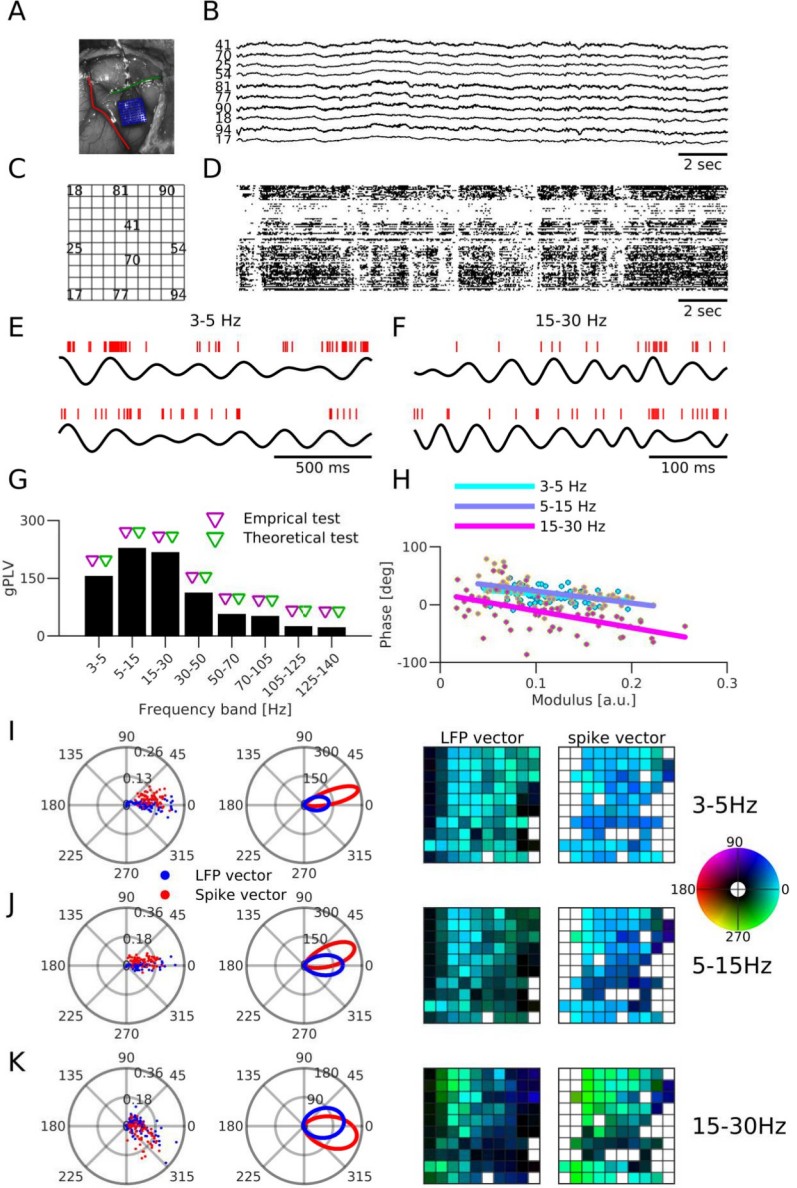

**Fig 8. Application to electrophysiological recordings in non-human primate PFC. (A)** Location of the Utah array, anterior to the arcuate sulcus (red line) and inferior to the principal sulcus (green line). **(B)** Broadband trace of the recorded LFP (from the recording channels indicated in C). **(C)** Utah array spatial map identifying channel IDs shown in B. **(D)** Spike rasters for all recorded neurons. **(E-F)** Example spike trains (red bars) and filtered LFP (black traces) in the frequency ranges (E) 3–5 Hz and (F) 15–30 Hz. **(G)** gPLV values. Triangles indicate the significance assessed based on surrogate (blue triangles) and analytical test (red triangles) tests. **(H)** Phase of spike vector coefficients as a function of its modulus for the frequencies indicated in the legend (one dot per coefficient, continuous lines indicate linear regression). **(I-K)** LFP and spike vectors for frequency (I) 3–5 Hz, (J) 5–15 Hz, and (K) 15–30 Hz. The first column depicts the LFP (blue dots) and spike (red dots) in the complex plane. The second column depicts the fitted von Mises distribution to phase of LFP and spike vectors. Third and forth columns respectively represent the spatial distribution of phase of LFP and spike vectors values on the array (see C). White pixels in the third column (LFP vector) indicate the recording channels that were not used in the recording and in the fourth column (spike vector), white pixels indicate the recording channels with insufficient number of spikes (multiunit activity with a minimum of 5 Hz firing). In the last two columns, colors are represented in HSV mode, in which a complex number ($re^{i\phi}$) is represented by hue and brightness of a pixel. The hue of a pixel indicates the phase ($\phi$) and the brightness of a pixel indicates the modulus ($r$). The colorbar is depicted on the right. Related supplementary Figure: S7 Fig, Analysis of PFC Utah array data.

*preparation and intracortical recordings*, and multi-unit activity with a minimum of 5 Hz firing rate was used. Recorded signals are exemplified in Fig 8B–8F. Exemplary LFP traces are illustrated in Fig 8B. Each trace is recorded from the location specified in Fig 8C. Spike trains are also displayed in Fig 8D (for the same epoch used in Fig 8B). As the analysis is performed in band-limited frequency ranges, we also exemplified band-passed LFP signals (together with spikes) in Fig 8E and 8F. The dataset consisted of 200 trials of visual stimulation (10 sec) and inter-trials (10 sec) each 20 sec.

Computing GPLA in different frequency bands revealed that the strongest coupling was in the alpha range (5–15Hz) (Fig 8G). Furthermore, we assessed the significance of coupling with both surrogate and analytical tests (see Materials and methods, section *Significance assessment of gPLV*). GPLA above 50 or 60 Hz should be considered with caution, as in high frequencies the spike-LFP relationships may be affected by the contamination of high frequency LFP bands by spike waveforms of units recorded in the same channel [97, 98]. This may bias spike-LFP coupling towards the specific relation between the spiking of those specific units and the surrounding field, instead of capturing the relation of the underlying population rate to this field, as assumed in neural field models.

Fig 8I–8K further shows the spike and LFP vectors for the three frequencies with the largest coupling according to their gPLVs (for other frequencies, see S7 Fig). Representing spike and LFP vectors in the complex plane (Fig 8I–8K first column), suggests that the relative phases of spike and LFP vectors are different across these three frequencies. To demonstrate the difference more clearly, we fit von Mises distributions to the pooled phase of all coefficients of the vectors (Fig 8I–8K second column). The sign of the spike-LFP phase differences changes across frequencies, with spikes ahead of time with respect to LFP in low frequency, while lagging at higher frequencies. This behavior is similar to the above analysis of strongly recurrent neural field model (Fig 7G), when EPSP is taken as an LFP proxy.

The spatial mappings of the LFP and spike vectors on the Utah array (Fig 8I–8K, third and fourth column) also demonstrate a spatial structure in the modulus and phase of the LFP and spike vectors, revealing localized regions with stronger participation in the locking, in particular in the beta range 15–30 Hz (green pixels at the middle-top and -bottom in Fig 8K, fourth column). We hypothesize this is due to a higher activation of spatially localized populations, as supported by anatomical studies of the PFC [99, 100] and electrophysiological [101] studies. Notably, capturing this aspect of the circuitry from the neural data based uni-variate phase locking analysis relies on finding a suitable choice of LFP reference channel, which is typically challenging (see S8 Fig, for comparison of multi-variate analysis and examples of uni-variate based on two different choices of reference channel).

Furthermore, in the alpha band (5–15 Hz), exhibiting the strongest coupling between spike and LFP, the spike vector coefficients' moduli are significantly negatively correlated with their phase (Fig 8H, $p < 10^{-6}$, F-test on the linear regression model; $N = 66$). Interestingly, we observe again a similar behavior in the above neural field simulation with strong recurrent inhibition, but not in the simulation with weak recurrent inhibition (Fig 7D). Notably, the result of a similar analysis based on uni-variate phase locking analysis leads to a profile incompatible with our conclusion based on neural field simulation (see S9 Fig).

Overall, these results suggest a neural field with excitatory horizontal connections and strong local recurrent inhibition as a plausible model for the recorded prefrontal circuits, in line with what has been suggested by previous modeling work [96, 102]. This analysis illustrates how GPLA can support the mechanistic understanding of high-dimensional experimental recordings.

## Discussion

In spite of the relevance of spike-field relationships for assaying coordination mechanisms in brain networks [19, 20, 22–24], they are still not systematically investigated in the context of highly multivariate recordings. Potential reasons could be the lack of multivariate methodologies for investigating such coupling beyond a single pair of spiking unit and a LFP channel, and interpretability challenges.

In this study, we developed Generalized Phase Locking Analysis (GPLA) as—to the best of our knowledge—the first *multivariate* method with *demonstrated biophysically interpretability* for investigating the coupling between spatially distributed spiking and LFP activity. GPLA summarizes the coupling between multiple LFP spatio-temporal patterns and multiple spiking units in a concise way. At a given frequency, the spike and LFP vectors represent the dominant LFP and spiking spatio-temporal distribution, while the generalized Phase Locking Value (gPLV) characterizes the strength of the coupling between LFP and spike patterns. Some of the conclusions we draw based on GPLA may to some extent also be achievable with univariate techniques, *but in contrast to GPLA, this typically requires ad hoc decisions or guiding univariate methods with considerable amount of prior knowledge on the structure under study*. For instance, univariate techniques can be used for analyses we described in Fig 6E and 6H, provided a suitable LFP reference channel is used to assess the coupling of all recorded units. Choosing such channel is not trivial unless it is justified, for example, by prior knowledge on the hippocampal circuitry. Even in such case, prior knowledge may not reflect accurately the properties of the recordings and bias the analysis. An arbitrary choice of reference channel will not faithfully reflect the dominant coherent activity with units primarily synchronize. Certainly, such caveats are even more pronounced when investigating structures with less prior knowledge and with recording techniques yielding a larger number of channels.

We demonstrated that GPLA's outcome features, such as the overall spike-LFP phase shift, the phase shift between different cell types (excitatory and inhibitory), and the spatial phase gradients, provide information about the overall organization of the recorded structure that are not easily quantifiable with simpler measurements.

First, application to realistic simulations of hippocampal SWR revealed various characteristics of hippocampal circuitry with minimal prior knowledge. Second, in order to better interpret spike and LFP vectors' spatial distribution, we also simulated spatially extended neural field models and demonstrated that phase gradients of spike and LFP vectors in these neural field models reflect properties of the underlying microcircuit connectivity (such as the strength of recurrent interactions). Finally, the application of GPLA to experimental recordings suggests a global coupling between spiking activity and LFP traveling wave in vlPFC in line with our simulations of a neural field endowed with strong recurrent inhibition.

Statistical properties of gPLV were investigated to develop an empirical and theoretical framework for assessing the significance of coupling. The theoretical statistical test built upon Random Matrix Theory [59] makes the method applicable to high dimensional electrophysiology data with low run-time complexity, which is important for modern probes such as Neuropixel, featuring 960 recording sites [40]. In contrast, conventional statistical testing procedures based on the generation of surrogate data become computationally expensive as the number of recorded neurons increases.

### Comparison to existing approaches

To the best of our knowledge, there are very few studies that include the information of multiple LFP channels *and* multiple spiking units for investigating spike-LFP interactions. In

particular, among approaches exploiting multiple LFP channels, none fully exploit the statistical relation between spiking activity recorded from multiple sites.

The Spike-Triggered Average (STA) of LFP is one of the common multivariate technique for characterizing spike-LFP relationship [103, 104]. It has moreover been interpreted as a measure of functional connectivity [104] (but also see [105] and [106]). Although STA can exploit multivariate LFP signals, it can only be computed based on a single spike train, thereby ignoring the information provided by the remaining units. Similarly, even sophisticated extensions of spike-triggered averaging of LFP [91] still rely on the information of individual spiking units. In a similar vein, the study of [107], which showed that the probability of spiking can be statistically related to the LFP phase in multiple distant regions, was also limited to spiking units taken individually.

This appears clearly as a limitation, because statistical relationships between the spiking activity of different units, such as lags between the activity of different types of neurons (e. g. excitatory and inhibitory neurons) [108] can inform us about the organization of the neural circuit. Notably, this is supported by our simulations and previous experimental work [85].

Apart from works that specifically target spike-field coupling, a body of methodological studies by *van der Meij* and colleagues use the idea of extracting a dominant frequency coupling structure with dimensionality reduction techniques [52–54], akin to GPLA's principle. In spite of the similarities between these methodologies from a data analysis perspective, GPLA-based investigation of spike-LFP coupling further leads to a biophysical interpretations in terms of underlying circuit properties, while this key question is left unaddressed by other approaches. This is due to the ability of GPLA to allow both dimensionality reduction (of experimental recordings) and model reduction (of neural field model) such that the outcome of both reductions can be related.

## Limitations and potential extensions

One limitation of GPLA is that it considers the underlying network dynamics to be fixed for the analyzed data. Although the use of GPLA on simulation of Hippocampal Sharp Wave-Ripples demonstrates that an application on even such transient and aperiodic signals is insightful (Fig 6), but certainly due to the non-stationarity of neural dynamics, the time-resolved analysis of spike-LFP data (that likely required further methodological development) may improve our understanding of the underlying processes. As an alternative, it is however possible to apply the present methodology to portions of recordings containing identified transient phenomena, such as hippocampal Sharp Wave-Ripples, that are likely key to understand brain function [109, 110]. For example, as LFPs result from the superposition of electric potentials from multiple sources and can capture various *coordinated or cooperative* phenomena, LFP decomposition techniques can temporally isolate these epochs of coordinated activity and application of GPLA to these epochs can characterize how each neuron is participating in the collective activity and/or to what degree, it is coupled to the larger-scale dynamics.

Another limitation comes from the nature of SVD, leading to orthogonal singular vectors that may or may not capture the properties of distinct physiological processes. We therefore mostly limited the scope of this study to capturing the dominant coupling between spikes and LFP, reflected in the largest SV and corresponding vectors. However, as simulations used in Fig 4 demonstrate in simple cases, the number of significant SVs may correctly identify the number of neuronal population coupled to different rhythms. In general, the amount of information neglected by limiting the analysis to the largest singular value, highly depends on the settings under study. We have demonstrated a variety of them in this manuscript. For instance, the simulation used in Fig 3 exemplifies a small loss, and for Fig 6 a large one. Certainly, more

quantitative approaches can also be taken, for instance, by quantifying this loss by the ratio of the largest singular value to the sum of all singular values.

Furthermore, GPLA can also be improved by exploiting a better univariate estimation method. Various novel methodologies for assessing pairwise spike-field coupling have been developed in recent years [34–38, 111, 112] each providing some improvements over classical measures such as PLV. For instance, [38] proposed a bias-free estimation of spike-LFP coupling in the low firing rate regime. Replacing the coefficients of coupling matrix (Eq 17) with these improved pairwise estimates may bring those benefits to GPLA as well. Nevertheless, the pairwise estimate used in the present paper has the benefit of yielding well behaved statistical properties as the number of recording channels gets large, allowing to quickly assess the significance of the coupling using Random Matrix Theory. Alternative pairwise coupling estimates would likely need to be adapted in order to preserve the statistical benefits of our approach. This typically requires calculating the asymptotic distribution of the coupling statistics and devising and appropriate normalization thereafter. In case the new coupling measures are not adaptable to the analytical test, the surrogate-based test remains applicable at the expense of heavier computational costs.

## Neuroscientific interpretation of GPLA

Due to the complexity of the structure and dynamics of spatially extended neural networks, interpreting the outcome of statistically sound approaches such as GPLA in terms of biological mechanisms remains challenging. Thanks to the analysis of neural mass/field models, we could link several features of GPLA to a mechanistic interpretation. First, we applied this strategy to simulations from a biophysically realistic model of hippocampal ripples in order to use a system for which the underlying mechanism are well understood, but more complex than the neural field models used to interpret GPLA results. Despite the discrepancy between models, this showed that increasing the complexity of neural mass models using properties that are qualitatively in line with the key ground truth underlying mechanisms (e. g. inhibitory synaptic delays), allowed reproducing qualitatively GPLA results of these simulations, making the approach interpretable. This allowed in particular (1) to relate the LFP vector to the laminar distribution of field potential generated by current dipoles, (2) to link the phases of the spike vector to cell types and recurrent I-I dynamics.

Next, we used neural field simulations in order to find interpretations of GPLA characteristics that can be exploited in the context of cortical electrode array recordings. This is an important step as the mechanisms underlying spatio-temporal phenomena observed *in vivo* remain largely elusive. While keeping the complexity of these models minimal (using exponentially decaying horizontal excitation and local inhibition), we could already observe that altering the microcircuit structure resulted in interpretable qualitative modifications of GPLA's outcome, in particular regarding the phase gradients of spike and LFP vectors across the array. Finally, our analysis of Utah array recordings suggests the key GPLA features exhibited in simulation can also be estimated in real data and provide insights into the underlying organization of the recorded circuits.

As mentioned when introducing the concept of biophysical interpretability, the reliability of mechanistic interpretations drawn from GPLA crucially depends on the ability of the reduced biophysical models that we use to approximate key ground truth mechanisms underlying the data. Although no absolute guarantees can be provided, we showed in two sets of simulated data that the linearized neural field approximations provided qualitative insights in line with ground truth mechanisms, which were based on more complex (notably non-linear) models. Overall, the simple rate models we investigated have the benefit of lending themselves

to approximate analytical treatment, providing direct insights into the role played by network parameters in GPLA characteristics. Neural mass modeling has of course inherent limitations due to approximating local population activity by their mean rate, such as their typical inability to account for synchronization of spike times. However, multiple refinements of these models have been developed and offer potential for improving biological realism. Notably, next generation neural mass models are able to capture event-related synchronization between neurons [113, 114] and can incorporate the dynamics of intrinsic currents that are key to modeling complex phenomena empirically observed, such as spindle oscillations [115]. In addition, neural field models can be improved in light of the knowledge about the horizontal connectivity of the structure, which may not be monotonous (for example see recent findings on non-monotonous correlation structure in V1 [116] and PFC [101]), and heterogeneous [117].

More generally, a mechanistic interpretation of GPLA results in a given structure strongly relies on the accuracy of the assumptions made to perform analytical and/or computational modeling. One aspect that entails limitations is the linear response theory on which we base our interpretations in the present work. Linearization is typically justified for a stable system exhibiting low amplitude fluctuations around its equilibrium point. However, non-linear model reduction techniques such as the Galerkin method [118] allows to extend low dimensional, interpretable approximations of high-dimensional systems to more general settings.

The investigation of more complex models will benefit from incorporating systematic parameter estimation approaches, taking inspiration from inference techniques that have been developed for modeling the activity of one or several neurons [119, 120] and combining them with the present model and dimensionality reduction approaches to ensure tractable estimation.

Ultimately, our results support the relevance of GPLA for studying distributed information processing in higher-tier cortical areas such as PFC and hippocampus, where spike-LFP interactions have proven key to elucidating the neural basis of cognitive functions such as working memory [121, 122], memory consolidation and spatial navigation [123, 124]. This approach is likely to provide further insights about coordination mechanisms by shifting the focus from properties of individual units to characteristics of spatially extended networks taken as a whole.

## Materials and methods

### Ethics statement

The neural data used in this study were recorded from the ventrolateral prefrontal cortex (vlPFC) of one anaesthetised adult, male rhesus monkey (macaca mulatta) by using Utah microelectrode arrays [Blackrock Microsystems [125]] (more details on these experiments are provided in a previous study exploiting this data by [101]). All experiments were approved by the local government authorities (Regierungspräsidium, Tübingen, Baden-Württemberg, Germany), and were in full compliance with the guidelines of the European Community (EUVD 86/609/EEC) for the care and use of laboratory animals.

### GPLA for electrophysiology data

GPLA proceeds in several steps: preprocessing of multi-channel LFP signals, construction of the coupling matrix, and its low-rank approximation. Finally, parameters of this low-rank approximation are standardized following specific normalization conventions allowing their easy interpretation and comparison. These steps are described in the following subsections.

**LFP pre-processing.** Prior to computing couplings, the LFP signal is pre-processed, first by filtering in the frequency band of interest. The choice of the filter bandwidth for the purpose

of extracting the instantaneous phase or analytic signal in a particular band is subjected to a trade-off. On one hand, the signal requires a narrow enough band-pass filtering to provide us a proper estimate of the phases [58]. On the other hand, the filtered signal should preserve the temporal dynamics in the frequency of interest. The second step extracts the analytical signal using the Hilbert transform, resulting in a complex-valued signal containing both the amplitude and phase of LFP. In the optional third step (see section *Necessity of whitening and post-processing*), we whiten the LFPs. We need to decorrelate LFP signals recorded in different channels by applying a whitening operator. Whitening is only necessary to be able to use tools from Random Matrix Theory [62] for the purpose of statistical analysis, otherwise generalized phase locking value, spike and LFP vectors can all be calculated in the original channel space. In both cases, GPLA outputs can be interpreted in the channel space (by inverting the whitening operation if it has been applied). For more detail on the rationale for the inclusion of the whitening step, see section *Analytical test* and [59].

We consider LFPs and spiking units are recorded repeatedly over $K$ trials, and each trial has length $T$ (number of time-points). We represent LFPs of trial $k$ by $L^{(k)}$, which is a $(n_c \times T)$ matrix, where $n_c$ is the number of LFP recording channels. To simplify the notations, by $L^{(k)}$ we refer to analytical signals, i. e. band-passed in a particular frequency range and Hilbert transformed signals. We denote the collection of $N_m^{(k)}$ spike times of unit $m$ at trial $k$ by $\{t_j^{m,(k)}\}_{j=1...N_m^{(k)}}$ ($\{t_j^{m,(k)}\}$ contains the time-point indices of the LFP data for which spikes occur).

We introduce a *reduced-ranked* whitening operator which is a modified version of the conventional whitening that decorrelates the data, in this case, LFP signals. We customized this procedure in order to accommodate GPLA's needs, i. e. (1) avoid over-amplification of noise components of LFP (which are reflected in smaller eigenvalues of LFP covariance matrix) in the whitening operator, and (2) eliminate factors of variability that are not consistent across trials.

In our *reduced-ranked* whitening, we first reduce the rank of the LFP covariance matrix, by truncating the eigenvalue decomposition of LFP covariance matrix. We choose the number of components such that 99% of variance is explained with the reduced rank covariance matrix. In order to find the number of components that account for 99% of the total variance of the LFP covariance matrix, we concatenate LFPs of all trials into a larger $n_c \times KT$ matrix, denoted by $L$ and compute the eigenvalue decomposition of the covariance matrix,

$$\text{Cov}(L) = \frac{1}{T}LL^H, \tag{10}$$

where $.^H$ indicates the transpose complex conjugate (should be noted that, analytical signal $L$, is a complex-valued matrix). We denote the number of components needed to explain 99% of variance of LFP covariance matrix by $n_c^{eff}$. We find the reduced number of components, $n_c^{eff}$, based on all trials, and we use $n_c^{eff}$ to define the whitening operator of individual trials. The reduced rank single-trial LFP covariance matrix is denoted by $\text{Cov}^{red}(L^{(k)})$, and computed as follows,

$$\text{Cov}^{red}(L^{(k)}) = \sum_{p=1}^{n_c^{eff}} \lambda_p^{(k)} x_p^{(k)} (x_p^{(k)})^H, \tag{11}$$

where $\lambda_k^{(k)}$ and $x_k^{(k)}$ respectively denote the eigenvalue and eigenvectors of the LFP covariance matrix of trial $k$. We denote the whitened LFP of trial $k$ by $L_w^{(k)}$, and compute it as follows,

$$L_w^{(k)} = (\Lambda^{(k)})^{\frac{-1}{2}} (X^{(k)})^H L^{(k)}, \tag{12}$$

where $\Lambda^{(k)}$ is a $n_c^{eff} \times n_c^{eff}$ diagonal matrix containing the eigenvalues of the above single-trial reduced rank LFP covariance matrix, and $X^{(k)}$ is a $n_c \times n_c^{eff}$ matrix containing the eigenvectors $x_k^{(k)}$.

**Coupling matrix.** Given the spike times of a single spike train $\{t_j^{(k)}\}_{j=1\ldots N^{(k)}}$ and $L_w^{(k)}$ a single channel pre-processed LFP analytic signal (as explained in section *LFP pre-processing*) and its phase $\phi\ (= \angle L)$, the conventional measure of spike-LFP coupling, Phase Locking Value (PLV), defined as follows:

$$PLV = \frac{1}{N^{tot}} \sum_{k=1}^{K} \sum_{j=1}^{N^{(k)}} \exp\left( \boldsymbol{i}\phi_{t_j^{(k)}}^{(k)} \right) , \qquad (13)$$

where, $\boldsymbol{i}$ is the imaginary unit ($\boldsymbol{i}^2 = -1$), and $N^{(k)}$ is the number of spikes occurring during the trial $k$, $N_{tot}$ is the total number of spikes occurred across all trials, i. e.

$$N^{tot} = \sum_{k=1}^{K} N^{(k)} . \qquad (14)$$

In addition to PLV, we introduce a similar coupling statistics, denoted by $c$,

$$c = \frac{1}{\sqrt{N^{tot}}} \sum_{k=1}^{K} \sum_{j=1}^{N^{(k)}} L_{t_j^{(k)}}^{(k)} , \qquad (15)$$

to be used when the theoretical significance test is intended to be used (see section *Analytical test*) (for summary on type of normalization used in different figures see Table 1). The coupling statistics $c$ is different from PLV in two ways. First, in PLV only the phase information from the continuous signal is used, while for $c$, we use both the phase and amplitude of the LFP signal. This is motivated by evidence that inclusion of the amplitude can improve the coupling measure [126, 127] by weighting the contribution of spikes in the coupling measure by the LFP amplitude at the correspond spike time, as well as by theoretical considerations (see section *Analytical test* for more details). The second difference is, for $c$ we have normalization by square root of the number of spikes rather the number of spikes (division by $\sqrt{N^{tot}}$ in Eq 15 versus $N^{tot}$ in Eq 13). Basically, a scaling by $\sqrt{N^{tot}}$ is needed to normalize the variance of entires of the coupling matrix to 1, in order to be able to use tools from Random Matrix Theory [62] (see [59] for more details).

A multivariate generalization of the coupling statistics, could be achieved by collecting the coupling statistics between all spiking units and LFP signals. Given spike times $\{t_j^{m,(k)}\}_{j=1\ldots N_m^{(k)}}$, $\phi_w^{(k)}$ LFP phase, and $L_w^{(k)}$ the analytical LFP, we can define the coupling matrix $\boldsymbol{C}$, based on PLV (Eq 13, also similar to [82]) as follows,

$$(\boldsymbol{C})_{n,m} = \frac{1}{N_m^{tot}} \sum_{k=1}^{K} \sum_{j=1}^{N_m^{(k)}} \exp\left( \boldsymbol{i}(\phi^{(k)})_{n,t_j^{(k)}} \right) , \qquad (16)$$

or based on $c$ (Eq 15),

$$(\boldsymbol{C})_{n,m} = \frac{1}{\sqrt{N_m^{tot}}} \sum_{k=1}^{K} \sum_{j=1}^{N_m^{(k)}} (L^{(k)})_{n,t_j^{(k)}} , \qquad (17)$$

where $m$, $j$ and $n$ respectively indicate the index of spiking unit, index of spike time and index

of LFP channel and $N_m$ refers to number of spikes recorded in spiking unit $m$. Readers can also refer to [59, Section 4] for a different formulation.

Let $n_c$ and $n_s$ be the number of LFP channels and number of spiking units, respectively, $C$ is thus a $n_c \times n_s$ complex-valued matrix (or $n_c^{eff} \times n_s$ if whitening is applied). As $n_c$ (or $n_c^{eff}$) and $n_s$ are not necessarily equal in electrophysiological datasets, the coupling matrix is not square in general.

Our coupling matrix is thus designed as a multivariate generalization of univariate coupling measures in order to capture the overall synchronization between the spiking activity and the phase of a global oscillatory dynamics in a given frequency band.

**Low rank decomposition.**   Each column of the coupling matrix $C$ has a common spiking unit whose locking is computed with respect to different LFP channels (called LFP vectors). Conversely, each row collects the phase locking values of all spiking channels to a common LFP reference channel. In order to achieve a compact and interpretable representation of this high dimensional object, we compute the Singular Value Decomposition (SVD) of the coupling matrix of the form

$$C \quad = UDV^H \quad = \sum_{k=1}^{p} d_k u_k v_k^H , \qquad (18)$$

where $(d_k)$ is a tuple of positive scalars, the singular values (SV), listed in decreasing order. The complex valued vectors $u_k$ and $v_k$ are, respectively, the $n_c/n_c^{eff}$- and $n_s$-dimensional singular vectors associated to a given SV $d_k$. One important property of SVD is that keeping only the first term in Eq (18), with SV $d_1$, achieves the best rank-one approximation of the matrix, $C \approx d_1 u_1 v_1^H$, in the least square sense [128, Theorem 7.29].

**Post-processing.**   In order to make the outputs of GPLA interpretable, we introduce a few post-processing steps. An unwhitening and rescaling procedure is introduced to reverse some normalization discussed in previous sections *LFP pre-processing*, Coupling matrix, and Low rank decomposition, and a rotational transformation is introduced in order to represent the singular vectors in a more interpretable fashion.

*Representation of singular vectors:* Following the conventional mathematical representation of SVD in Eq 3, $U$ and $V$ are unitary matrices i. e. $U^H U = I$ and $V^H V = I$. This implies that all singular vectors are unit norm, and all the information regarding the strength of coupling is absorbed in the singular values on the diagonal matrix $D$. As explained in main text (see sections *Reduction of complex models based on linear response theory* and *Generalizing SFC to the multivariate setting*), the relative magnitude and phase of singular vectors coefficients can be used to interpret the *relative* contribution of individual LFP channel and individual spiking unit to the coordinated pattern captured by the largest singular value.

We can summarize the coupling matrix with three quantities:

$$C \sim (gPLV).v_{LFP}v_{spike}^H . \qquad (19)$$

However the coefficient of both singular vectors can be rotated of the same arbitrary angle in the complex plane, as the rotation transformation in the complex plane does not change the SVD factorization, i. e.

$$udv^H = udv^H e^{-i\theta_0} e^{i\theta_0} = e^{-i\theta_0} ud(e^{-i\theta_0}v)^H . \qquad (20)$$

We exploit this free parameter to make the GPLA more neuroscientifically interpretable by shifting the phase of both spike and LFP vectors with $-\overline{\phi_{LFP}}$, where $\overline{\phi_{LFP}}$ and $\overline{\phi_{spike}}$ are the

average spike and LFP phases, defined as,

$$\overline{\phi_{LFP}} = \angle \sum_{i=1}^{n_c} (v_{LFP})_i \, , \tag{21}$$

$$\overline{\phi_{spike}} = \angle \sum_{i=1}^{n_u} (v_{spike})_i \, . \tag{22}$$

The rationale behind it is to center the coefficient of the rotated LFP vector $(\widetilde{v_{LFP}} = v_{LFP} e^{-i\overline{\phi_{LFP}}})$ around zero phase in the complex plane and the rotated spike vector,

$$\widetilde{v_{spike}} = v_{spike} e^{-i\overline{\phi_{LFP}}} \tag{23}$$

preserves the angular difference of $\Phi_d$ of the spikes with respect to the LFP, defined as

$$\Phi_d = \overline{\phi_{LFP}} - \overline{\phi_{spike}} \, . \tag{24}$$

With this chosen convention, we obtain the final GPLA factorization

$$C \sim (gPLV).\widetilde{v_{LFP}}\widetilde{v_{spike}}^H \, . \tag{25}$$

We can also apply the phase difference between average LFP and spike vectors ($\Phi_d$) to gPLV as it can summarize the overall phase shift between LFP and spikes. Given that gPLV is always a real positive value, by this convention, we add an extra information to gPLV.

We thus define a *complex gPLV* ($\widetilde{gPLV} = gPLV e^{-i\Phi_d}$) whose magnitude indicates the coupling strength between spikes and LFPs as in phase locking value (PLV) and its angle indicates the overall phase difference between spiking activity and LFP which is similar to locking phase in classical univariate phase locking analysis. This is an arbitrary choice to some degree, nevertheless it allows to interpret the GPLA output similarly to classical univariate phase locking analysis. Needless to mention, when the magnitude of gPLV is small, this overall phase difference is not meaningful (similar to the case where PLV is small, the locking phase is not meaningful).

*Unwhitening:* As discussed in section *LFP pre-processing*, due to theoretical considerations, and in particular for applicability of our analytical significance test (see Significance assessment of gPLV), we whiten the LFPs prior to any other processing. In order to retrieve the original structure of the LFP i. e. retrieve all the correlations that were present in the original LFP signals but was diminished by the whitening, we need to "revert" the whitening i. e. unwhiten the LFP vector resulting from GPLA. This can be achieved by computing the unwhitening operator $W^{-1}$ and apply it to the LFP vector,

$$v_{LFP}^{unwhiten} = W^{-1} v_{LFP} \, . \tag{26}$$

In order to find this operator, we first concatenate whitened LFPs of all trials (resulting from Eq 12) into a larger matrix $L_w$ ($n_c^{eff} \times KT$). Then we estimate $W^{-1}$ by using a linear regression with unwhitned and whitened LFPs ($W^{-1}$ is the $n_c \times n_c^{eff}$ matrix of coefficient for regression).

*Rescaling:* As introduced in Eq 15, the coefficients of the coupling matrix are normalized by the square root of the number of spikes. This choice of normalization is different from the one used in conventional PLV (Eq 13). This will lead to inhomogeneous weighting of spiking units according to their variability of their firing rate. We "revert" this weighting later on by dividing

the spike vector by the square root of number of spikes,

$$v_{Spike}^{rescaled} = v_{Spike} \oslash \vec{N} \,, \tag{27}$$

where $\oslash$ is the (entrywise) Hadamard division and $\vec{N} = \{N_m^{tot}\}_{m=1,\dots,n_s}$, which is a vector consisting of total spike counts (similar to Eq 14) of all the neurons (indexed by $m$) used in GPLA. Furthermore, to preserve the original norm of the spike vector (unit magnitude), we also need to normalize the spike vector by its norm,

$$v_{Spike}^{final} = \frac{v_{Spike}^{rescaled}}{\| v_{Spike}^{rescaled} \|} \,. \tag{28}$$

**Necessity of whitening and post-processing.** The whitening (and the subsequent post-processing) is necessary to have the advantage of applicability of the analytical significance test. LFPs are typically very correlated signals, leading to strong statistical dependencies between the coefficients of the estimated coupling matrix $C$, which affects the statistics of the singular values (and consequently gPLV). Whitening removes correlations before computing spike-LFP coupling. However, if statistical testing based on surrogate data is intended, it is possible to skip the whitening step and proceed directly with constructing the coupling matrix and low rank estimation (see Fig 3). In that case, entries of the coupling can be filled by conventional PLVs (see Eq 16), or other choices of spike-LFP coupling measures [34–38, 111, 112] (also see the section *Limitations and potential extensions* for further elaboration). In this case, whitening of the LFP can be skipped and subsequent "Unwhitening and rescaling" discussed in section *Post-processing* is not necessary anymore.

**Optional normalization for gPLV.** As gPLV is a singular value of a matrix, it grows with the dimensions of the coupling matrix. This makes the comparison of gPLV resulting from different datasets difficult. For instance, assume the hypothetical situation of having two datasets recorded from two homogeneous populations of neurons, if the strength of coupling is the same in two populations, the populations with a larger amount of recorded neurons (therefore larger dimension of the coupling matrix) will have larger gPLV. Certainly, this can be misleading for investigating the spike-LFP coupling with GPLA when datasets with variable number of spiking units and/or LFP channels. To overcome this issue, we suggest normalizing the gPLV to become independent of the size of the neural population (dimension of the coupling matrix) and the number of channels. When we consider the entries of coupling matrix, $C$, to be PLV (LFPs are not whitened and Eq 16 is used for constructing the coupling matrix), pairwise coupling static is bounded ($|PLV| \leq 1$). When all the entities of the coupling matrix $C$ attain their maximum value, gPLV will also gain the maximum possible value. Therefore, we can exploit it to normalize the gPLV. For a coupling matrix having maximum coupling for all pairs ($(C)_{n,m} = 1$ and $C$, a $n_c \times n_s$ matrix), then $gPLV_{max} = \sqrt{n_c n_s}$. Therefore, if we normalize the original gPLV by the maximum value it can achieve ($gPLV_{max} = \sqrt{n_c n_s}$, calculated is based on the dimensionality of the matrix $C$), then the gPLV will be bounded by 1 as well. Moreover, with this normalization, gPLV is also comparable to PLV (if we have a homogeneous population of neurons, otherwise these quantities are not comparable).

## Significance assessment of gPLV

In order to statistically assess the significance of the coupling between spikes and LFP based on gPLV, we develop a surrogate- and a Random Matrix Theory (RMT)-based statistical testing framework exposed in [59]. Hypothesis testing based on the generation of surrogate data is a

common method for significant assessment in neuroscience. Nevertheless, not only generating appropriate surrogate data can be challenging (for a review see [45]), but also computationally expensive. This motivates the development of an "analytical" test exploiting minimal computational resources.

**Surrogate-based test.** In contrast to univariate methods for which the distribution under a null hypothesis is more likely to be (possibly approximately) derived based on theoretical analysis (e.g., Rayleigh test for PLV [129, Chapter 4]), such approaches are usually unavailable in multi-variate settings (nevertheless, we have developed one for gPLV, see section *Analytical test*). Following a common alternative approach, we build the null distribution by generating many surrogate datasets [45]. The resulting gPLVs values forms an empirical $H_0$ distribution that can be used to compute the p-value for statistical assessment of the significance gPLV in the data. Importantly, the choice of appropriate surrogates according to characteristics of neural data is critical. For instance, generating surrogate data by shuffling inter-spike-intervals (ISI) is not an appropriate method when we have non-stationarity in firing rates [45].

In this work, we used an *interval*-jittering rather than a *spike-centered*-jittering (interval- and spike-centered-jittering are also known as hard and soft dithering, respectively), as the former was reported to be more reliable for detecting temporal structures in spike data [130]. We devised the two following spike-jittering-based methods for GPLA. We also verified the appropriateness of our jittering approach with various simulations (see the Results).

**Simple interval jitter.** Each surrogate dataset is generated by jittering all the spikes (from all neurons) with a particular jittering window (or dither width). In the interval jittering, per each spike, a new spike time is drawn within the jittering window around the spike. The timing of jittered spikes should be drawn from a uniform distribution. The size of the jittering window can be specified by the frequency wherein the spike-LFP coupling is being investigated. The smallest jittering window (or dither width) that can be used in order to destroy the temporal structure potentially exists in the range of frequency-of-interest. In the phase-locking analysis of electrophysiological data we usually extract the analytic signal or instantaneous phase of LFP by applying Hilbert transform on band-limited LFP signals [58]. The central frequency of the band-limited filter can be used for specifying the jittering window (or dither width), i. e. jittering window is the inverse of this central frequency.

**Group preserved jitter.** Similar to "simple interval jitter" we generate each surrogate dataset by relocating all the spikes within a window. For each surrogate data, we first divide the spike trains into equally-sized windows. Then we circularly shift the spike sequence within each window for all neurons together using a uniformly distributed time shift. Notably, we use a single random value for circular shifting of all neuron's spiking within the window. This size of this window should be chosen similar to the previous method ("simple interval jitter") i. e. based on the central frequency of the band-limited filter. The rationale behind this method of generation surrogate data is *relative* timing of the spikes could be associated to a large degree to the ansamble activity irrespective of the coupling to the LFP. Therefore, the relative timing of the spikes might not be impaired in the absence of coupling to global dynamics of the LFP. With "group preserved jittering" the relative timing is preserved and the coupling to the LFP is destroyed.

**Analytical test.** Challenges in generation of surrogate data [45] and considerable increase in the dimensionality of datasets [24, 40, 42, 43], suggest that deriving mathematically (asymptotic) properties of GPLA under the null hypotheses, as is done for univariate testing (e. g. Rayleigh test for PLV [129, Chapter 4]) is an interesting alternative.

In a companion work [59], by using martingale theory [131] we derive an asymptotic distribution for the entries of the coupling matrix in fairly general settings. Furthermore, by exploiting RMT [62] we can find a good approximation of the distribution of eigenvalues (or singular

values) of the coupling matrix in absence of coupling between spikes and LFPs. This provides a null hypothesis for the statistical testing of the largest eigenvalues (or singular values) of the coupling matrix, which corresponds to gPLV in our setting.

As mathematical details are described in [59, Theorem 2], we restrict ourselves to a brief explanation. When the LFP signal is whitened, and under a null hypothesis reflecting an *absence of coupling*, the coupling matrix which is constructed based on Eq 15, asymptotically converges to a matrix with i.i.d. complex standard normal coefficients [59 Theorem 3], and the Marchenko-Pastur (MP) law then provides an approximation of the distribution of its squared singular values [59, Theorem 3].

This law [64] has density

$$\frac{d\mu_{MP}}{dx}(x) = \begin{cases} \frac{1}{2\pi\alpha x}\sqrt{(b-x)(x-a)}, & a \leq x \leq b, \\ 0, & \text{otherwise}, \end{cases} \tag{29}$$

with $a = (1-\sqrt{\alpha})^2$ and $b = (1+\sqrt{\alpha})^2$ which are the upper and low bounds of the support of the distribution. Based on the these bounds we can define a significance threshold, $\theta_{DET}$, for the largest eigenvalue of hermitian matrix, $\boldsymbol{S} = \frac{K}{n_u}\boldsymbol{C}\boldsymbol{C}^H$:

$$\theta_{DET} = \left(1+\sqrt{\alpha}\right)^2. \tag{30}$$

The null hypothesis can be rejected if, the largest eigenvalue of $\boldsymbol{S}$ (denoted by $\ell_1$) is superior to the significance threshold:

$$\ell_1(\boldsymbol{S}_n) > \theta_{DET}. \tag{31}$$

Therefore, there is a significant coupling between the multi-channel spikes and LFPs, if

$$gPLV > \sqrt{n_u \theta_{DET}}. \tag{32}$$

As mentioned above, to be able to use the result of [59], we need to whiten the LFP signal first, as described in LFP pre-processing. Furthermore, satisfying this theorem requires the coupling matrix to be normalized appropriately based on the spike rate of each unit (as defined in Eq 17).

For computing $\alpha$ on neural data, the *reduced ranked* $n_c^{eff} < n_c$ entailed by the whitening procedure (see LFP pre-processing for more details), the *effective* dimensionality of the coupling matrix changes from $n_c \times n_u$ to $n_c^{eff} \times n_u$ (which depends on the spectral content of the LFP). This leads to a modification of Eq 30 as follows:

$$\theta_{DET} = \left(1+\sqrt{\alpha_{eff}}\right)^2, \tag{33}$$

where $\alpha_{eff} = n_c^{eff}/n_u$.

## Supporting information

**S1 Appendix. Contains method details and analytical developments.**
(PDF)

**S1 Fig. Use of EPSP as LFP proxy.** Difference between phase of excitatory and inhibitory neurons/populations based on GPLA and the excitatory and inhibitory populations in the MassAlpha neural mass model. In this simulation EPSP has been used for the LFP proxy.
(PDF)

**S2 Fig. GPLA of CA3 and CA1 activities.** For this analysis, CA1 and CA3 data were separately injected into GPLA. **(A)** Spike vectors represented in polar plots similar to Fig 6E, but for all frequencies (indicated on the left). **(B)** LFP vectors, similar to Fig 6D, but for all frequencies (indicated in legend in the bottom). **(C)** gPLV for different frequency ranges of LFPs, similar to Fig 6C. Triangles indicated the significance assessed based on empirical (blue triangles, with significance threshold of 0.05) and theoretical (red triangles) tests. (left) for CA3 and (right) for CA1.
(PDF)

**S3 Fig. Joint CA1-CA3 analysis of hippocampal SWRs.** For this analysis, CA1 and CA3 data were injected to GPLA together. **(A)** Spike vectors represented in polar plots similar to Fig 6E, but for all frequencies (indicated on the left). **(B)** LFP vectors, similar to Fig 6D, but for all frequencies (indicated in legend in the bottom). **(C)** gPLV for different frequency ranges of LFPs Fig 6C. Triangles indicated the significance assessed based on empirical (blue triangles, with significance threshold of 0.05) and theoretical (red triangles) tests.
(PDF)

**S4 Fig. GPLA vs. PLA comparison for hippocampal SWR simulation.** Similar to Fig 6D but based on uni-variate phase locking analysis (rather than multivariate GPLA). Each line depicts the phase locking value (PLV) for a fixed spiking units across all LFP channels. Colors indicate the frequency of filtered LFP.
(PDF)

**S5 Fig. Phase-modulus relation dependency on level of inhibition.** Related to Fig 7D. **(A)** Same as Fig 7C. for simulations at intermediate levels of recurrent inhibition. **(B)** Same as Fig 7D. for simulations at intermediate levels of recurrent inhibition. **(C)** Same as Fig 7E. for simulations at intermediate levels of recurrent inhibition. **(D)** Magnitude of phase modulus regression coefficient (rescaled by $180/\pi$ to have it in radians) as a function of imaginary part of $a$ derived from Eq 9. **(E)** Same as (A) for $Im[a]/Re[a]$ instead of $Im[a]$.
(PDF)

**S6 Fig. GPLA using IPSP as LFP proxy.** To be compared with Fig 7C–7E. **(A)** gPLV as a function of frequency for both models. **(B)** Phase of spike vector coefficients as a function of its modulus for the frequency band associated with maximum gPLV for each model (each dot one coefficient, and the continuous lines are plotted based on linear regression). **(C)** Shift between the averaged phase of spike vector and averaged phase of LFP vector, as a function of frequency.
(PDF)

**S7 Fig. Analysis of PFC Utah array data.** LFP and spike vectors for frequencies indicated on the right. First column depict the LFP (blue dots) and spike (red dots) in the complex plane. Second column depict fitted von Mises distribution to phase of LFP and spike vectors. Third and forth column respectively represnting phase of LFP and spike vectors which remapped to real configuration of electrodes on Utah array (see Fig 8C).
(PDF)

**S8 Fig. GPLA vs. PLA comparison for PFC Utah array data for revealing the spatial pattern of coupling.** Similar to Utah array maps in Fig 8I but based on uni-variate phase locking analysis (rather than multivariate GPLA). Panels in the first row depict the spatial distribution of phase locking value (PLV) or magnitude of the spike-field coupling on the array (see Fig 8C). Panels in the second row depict the spatial distribution of locking phase on the array. White pixels in all panels indicate the recording channels with insufficient number of spikes

(multiunit activity with a minimum of 5 Hz firing), as it was used in Fig 8I. The colorbars indicate the coupling strength in the first row; and locking phase in the second row. First column, depicts the results based on multivariate GPLA, and second and third column depicts the results based on uni-variate phase locking analysis, but for two different choices of LFP reference channel. The result from 'Example 2' is close to what is captured based on GPLA, however result from 'Example 1' does not, due to a lack of global coupling.
(PDF)

**S9 Fig. GPLA vs. PLA comparison for PFC Utah array data for characterizing the strength of recurrent inhibition in PFC circuits.** Similar to Fig 8H but based on uni-variate phase locking analysis (rather than multivariate GPLA). Each row corresponds to analysis in different frequency (the same frequencies used in Fig 8H), i. e., 3–5 Hz, 5–15 Hz, and 15–30 Hz, respectively, first, second and third row. First column indicates the results based on GPLA (notably pairwise coupling measure used here is exactly PLV), and imply the negative slope, similar to Fig 8H. The second and third columns demonstrate a similar analysis based on phase locking analysis, i. e., locking phase plotted versus strength of coupling (PLV) with two example of LFP reference channels (the same used in S8 Fig) Notably, none are compatible with our mean-field analysis (Fig 7). PLA is thus not conclusive about the strength of recurrent inhibition.
(PDF)

## Acknowledgments

We thank Britni Crocker for help with preprocessing of the data and spike sorting; Joachim Werner and Michael Schnabel for their excellent IT support; Andreas Tolias for help with the initial implantation's of the Utah arrays.

## Author Contributions

**Conceptualization:** Shervin Safavi, Theofanis I. Panagiotaropoulos, Michel Besserve.

**Data curation:** Shervin Safavi, Theofanis I. Panagiotaropoulos, Vishal Kapoor, Michel Besserve.

**Formal analysis:** Shervin Safavi, Michel Besserve.

**Funding acquisition:** Nikos K. Logothetis.

**Investigation:** Shervin Safavi, Theofanis I. Panagiotaropoulos, Vishal Kapoor, Michel Besserve.

**Methodology:** Shervin Safavi, Juan F. Ramirez-Villegas, Michel Besserve.

**Project administration:** Theofanis I. Panagiotaropoulos, Michel Besserve.

**Resources:** Nikos K. Logothetis.

**Software:** Shervin Safavi, Michel Besserve.

**Supervision:** Theofanis I. Panagiotaropoulos, Michel Besserve.

**Visualization:** Shervin Safavi, Michel Besserve.

**Writing – original draft:** Shervin Safavi, Michel Besserve.

**Writing – review & editing:** Shervin Safavi, Theofanis I. Panagiotaropoulos, Vishal Kapoor, Juan F. Ramirez-Villegas, Nikos K. Logothetis, Michel Besserve.

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
