## [Decision Letter · Decision Letter 0]

1 Aug 2022

Dear Dr. Besserve,

Thank you very much for submitting your manuscript "Uncovering the Organization of Neural Circuits with Generalized Phase Locking Analysis" for consideration at PLOS Computational Biology.

Finding good and available reviewers is just crazy at the moment. Given your response to the reviewers of the submission to ELife, and the very good quality of  Richard Gao's review, I decided to stick to one single review for this submission.

The main point to address regards the nature of the proposed method, as well as its applicability. This also reflects on the presentation of the work.

Please make sure that the code is in good shape and well documented in a third party public repository.

We cannot make any decision about publication until we have seen the revised manuscript and your response to the reviewers' comments. Your revised manuscript is also likely to be sent to reviewers for further evaluation.

Sincerely,

Daniele Marinazzo

Deputy Editor

PLOS Computational Biology

Reviewer's Responses to Questions

**Comments to the Authors:**

Reviewer #1: Apologies for the late review submission, and I don’t know if this is the intended behavior, but I wrote my review before reading the attached eLife reviews in order to not be biased towards existing opinions.

---

Summary:

In this paper, Safavi and colleagues present several contributions:

1. they proposal and implement a novel algorithm to measure multivariate spike-field coupling, termed gPLA, which is essentially taking the SVD of a complex-valued matrix consisting of each univariate pairwise (neuron x LFP channel) PLV estimate at a chosen frequency. The first (largest) singular value degree is taken as the total phase locking in the measured population / LFP channels, and the respective spike and LFP singular vectors denote the relative coupling degree and phase offset of that neuron / LFP channel.

2. they test the method on a simple simulation and demonstrate that gPLA is able to detect scenarios with homogeneous / clustered / graded variation in coupling to a 1D LFP signal, and that the method is robust to noise and outperforms standard univariate measures

3. they construct two statistical tests, one based on random matrix theory and the other on surrogate (jittered spiketime) analysis, to determine whether any detected coupling is statistically significant

4. they show analytically, in a simplified and linearized model of E-I network with external input, that the coupling function in the model reduces to a multiplicative scaling of the input, and that these coupling coefficients sampled at discrete spatial locations correspond to the spike and LFP singular vectors

5. they apply the method to a spiking neural network model of hippocampal SWR with HH neurons, as well as a more complex (less assumptions than in 4) neural field model, and show that gPLA outputs correspond to certain model parameters, and / or can be used for model comparison, and are consistent with known / implicated anatomical knowledge such as hippocampal cell-type specific phase locking or prefrontal connectivity motif

6. they apply the method to Utah array recordings from NHP PFC and find heterogeneous coupling frequency and phase across the recording sites, and that gPLA output qualitatively resembles that of the neural field model with strongly recurrent inhibition

From this series of observations, the authors argue that gPLA is a superior method for estimating multivariate spike-field coupling in a systematic fashion, and that since the analysis output can be used to distinguish or make inferences about model parameters from simulated data, that it can also be applied on real neural data to gain mechanistic (i.e., anatomical or physiological) insight, which has canonically been missing where SFC analyses are concerned other than broad references to communication through coherence.

---

Comments:

Overall, this paper is packed with a lot of interesting methodological innovations, theoretical / analytical insights, and analyses. In my opinion, the authors’ high-level goal of linking the family of SFC analyses to biologically meaningful quantities such as connectivity is of great interest to the systems neuroscience community, especially those concerned with theories and neural implementation of inter-areal communication. The specific gPLA method with the SVD implementation is novel as far as I know, and has the kind of insight that makes you wonder why someone else hasn’t applied SVD/PCA to the multivariate coupling matrix sooner. The analytical derivation is quite interesting, and the authors arrive at a neat result, though it requires quite a strong set of assumptions (this concern is expanded in full below). Lastly, they test their method on a series of models that increase in complexity to show that gPLA output can map to certain parameters of the models, or aid in model selection (e.g., low vs. high recurrent connectivity).

Despite the impressive collection of modeling and analysis results, I have reservations regarding the authors main claim, e.g., “…univariate SFC measurements […] remain blind to the role played by the overall organization of brain structures in shaping this coupling […]”, while “GPLA […] provide rich information to help uncover how biophysical properties shape network activity” (abstract L9 & L14). My main concern is that, while I think the method itself is innovative for performing SFC analysis in a naturally multivariate way, the claim that gPLA uncovers biophysical properties / mechanisms of neural circuits is not supported. As I summarized above, I believe the authors’ line of reasoning is that, since gPVA output corresponds analytically to the multiplicative transfer functions in the linearized network model, and qualitatively to various aspects of the spiking and neural field models, these insights then carry forward in a general way to real data as well. I don’t think this inference logically holds for the following reason: given simulated data (from a specific circuit model, spiking or neural mass), gPLA is one of many analyses one can run on the resulting spiking and LFP data. If the result of this analysis corresponds to some aspect of the generative model, it doesn’t necessarily hold that the same can be said about the real data if the generative model does not capture the necessary complexity and idiosyncrasies of the neural circuit in question (e.g., hippocampus vs. PFC vs. V1). Equally important, it doesn’t mean other methods cannot also provide the same mechanistic insight, especially when the “insight” in question is model selection between a small number of models, e.g., categorically low or high recurrent connectivity in the neural field model. While the authors acknowledge this in the discussions, e.g., referencing the fact that similar conclusions may be drawn with univariate PLV analyses, the broader point is that the inference is only possible because the space of models is limited a priori, which, I believe, is what does the heavy lifting in providing biophysical insight. To put it differently, for example in the introduction, the authors state that “SFC is typically restricted to a phenomenological description” (L45), but that has nothing to do with univariate SFC methods. In theory, one can go through the entire paper and replace gPLA with standard PLV and arrive at similar biophysical insights (in fact, it’s up to the authors to show that this is not the case), but only because a mechanistic model is proposed, not because gPLA is inherently different.

It’s possible that I’m putting too much emphasis on this point while the authors did not intend to imply that gPLA is somehow on higher epistemological ground than PLV. If that’s the case, I recommend clarifying in the writing to clearly distinguish the two different (and very valuable) contributions: the first being the multivariate SFC method, and the second being the model-driven framework of interpreting SFC-type analyses in general, and the demonstration of its utility on the various models, leading up to the NHP analysis. Furthermore, I believe more comparison is needed against standard PLV (or some other univariate metric) for the SWR and neural field models, to show that traditional methods cannot provide the same mechanistic conclusions / differentiate between different models. Simply mentioning the fact that univariate methods require more compute and some arbitrary decisions in choosing the reference LFP channel is NOT sufficient to discount existing methods, especially if similar insights can be extracted. Such comparisons would demonstrate how gPLA (and naturally analyzing coupling in a multivariate way) is preferable to univariate methods, while I believe more emphasis should be placed in the writing that the insights are gained from the combination of gPLA AND a model that will be area- and dataset-specific, and not a generic property of gPLA. Furthermore, if one takes this route, it would be preferable that each model is always paired with real data, which currently, only the neural field model does (loosely with the NHP data). I suppose the theoretical derivations centered on the reduced linear model does not require real data analysis (even though that would go towards justifying the assumptions made in the model), but the SWR analysis could be demonstrated on one of the many hippocampal datasets that’s openly available (e.g., on CRCNS).

On the other hand, if the authors do intend to suggest that gPLA somehow inherently allows more access to biophysical / mechanistic interpretations of the data (as is currently my reading of, e.g., L72-79, and L144-146), and is generalizable beyond the specific set of generative models they used in conjunction with the analysis method itself, then I have a number of concerns regarding the validity of such inference when strong assumptions / restrictions are made about the model:

- I believe a central piece of evidence that allows gPLA to make mechanistic links is the fact that the spike and LFP singular vectors correspond to the transfer functions in the linearized E-I neural mass model. While the analytical derivation is beautifully done, ultimately, this is an extremely impoverished model of the cortex, as it seems to me that the full set of assumptions essentially reduces the model to a linear network without recurrent inhibitory connections. And since the LFP is computed from the current fluctuations of the exc population, the model is basically a 1-population excitatory network driven by oscillatory external input, and inhibition can be ignored entirely. I think further discussion is warranted on if and whether the theoretical conclusions drawn from this model can be extended to more complex configurations; if so, how, and if not, does this undermine the validity of the analytical results (note that this is just about the derivations and analytical argument, as the later models increase in complexity in an empirical way but do not convincingly show the same analytical insight) .

- for the SWR model, it appears that gPLA reproduces experimental observations of laminar current profile and cell-type specific coupling phase in Fig 6A-D. However, it’s a bit unclear to me why the authors chose to add an alpha synapse / delay in the I-to-I connections, among many other possible model ingredients. By adding this component to the Mass2D model, MassAlpha qualitatively produces the biphasic shape of E-I phase shift across frequencies, but does this really imply that gPLA arrives at the “correct” mechanistic conclusion? Suppose some other biophysical detail was added to the model, such as the spatial distribution of synapses along the hippocampal layers, and that it also produces a similar shape as gPLA in Fig 6F, how would one differentiate between the competing hypotheses? While it’s promising that gPLA “with the appropriate simulations capture key characteristics of the underlying circuitry” (L451, again in L466), it also does not rule out the possibility that some other (univariate) SFC metric could do the same. Again, this does not take away from gPLA’s value as a multivariate analysis tool, but such a possibility would argue against the authors’ claim that it uniquely reveals mechanistic insight when combined with the appropriate model.

- similarly, for the neural field model, the authors essentially restrict the model space to two discrete possibilities: weak vs. strong recurrent inhibition. It is promising that gPLA output can distinguish between these two possibilities, but that does not imply that gPLA output provides mechanistic insight, especially in real data. It’s implied in the comparison with the real data that PFC connectivity is more consistent with the strongly recurrent model, but to what degree? At the minimum, the authors should vary inhibitory connectivity strength in a graded fashion and show that gPLA output scales parametrically, thereby providing more support for making an inference where, e.g., the slope of phase difference across frequencies corresponds to inhibitory connection density in a mechanistic way. But even then, the authors would have to explain why 1) no other model parameters would strongly influence this particular readout, 2) no other SFC metric (that they currently consider as phenomenological) can provide a readout of this model parameter even given the same circuit model, and 3) that the insight gained from the neural field model (with the restricted set of free parameters) is applicable to real data in a general way (i.e., if I applied the method to recordings from the visual cortex, would the same readout imply the same conclusion regarding circuit connectivity as for PFC? Why or why not?). And if not, these should all be discussed as limitations.

These are just a few specific examples that I believe could provide more support to the central point that gPLA provides access to mechanistic insight over existing (“phenomenological”) SFC metrics, but I hope my point is clear. To reiterate, I think the framework of pairing a (novel and interesting) data analysis tool with a mechanistic model that provides different hypothesis for how the empirically observed data could’ve been generated is an extremely important direction for systems and cognitive neuroscience. This framework (or philosophy) of computational modeling and analysis predates this particular paper, of course, but I commend the authors for taking such an approach, and I think this paper can be reorganized to emphasize that aspect more prominently. My main contention is regarding the point that, if I’m reading it correctly, it seems that the authors claim that gPLA inherently provides mechanistic insight in ways that other SFC methods cannot, independent of the specific mechanistic model and dataset used. At the very least, the argument should be that, given a SPECIFIC mechanistic model and dataset, a method that naturally accounts for multivariate data provides a better view into the underlying circuit parameters than a univariate method (which has to be shown).

A more general note about the presentation: currently, the paper reads a bit like a collection of nice but somewhat unrelated projects that use the same analysis method on vastly different computational models. It’s hard to say if there is a central point that these analyses all go towards supporting, or if there is a connection between the different model choices. If the intent is simply to demonstrate how gPLA may work with models of varying complexity to extract biological insight, then I would recommend explicitly stating so in the abstract or early in the introduction, and emphasize that the broad applicability of gPLA is a feature (but why that is not true of other SFC methods, even univariate ones). Perhaps it’s that gPLA depends on certain assumptions of spatial structure about the model / data-generating circuit, which is what allows it to extract that information? On the other hand, if there is a high level point that all the models support, I would recommend making that and the connection between them explicit (i.e., why they were chosen). I’m sorry that I don’t have a more concrete recommendation on the writing, but my feedback as a reader is that it was quite overwhelming to go through the manuscript even though I am familiar with most of the analysis and modeling techniques.

Richard Gao, PhD

University of Tuebingen

**Have the authors made all data and (if applicable) computational code underlying the findings in their manuscript fully available?**

Reviewer #1: Yes

PLOS authors have the option to publish the peer review history of their article (what does this mean?). If published, this will include your full peer review and any attached files.

Reviewer #1: **Yes: **Richard Gao

Figure Files:

Data Requirements:

Please note that, as a condition of publication, PLOS' data policy requires that you make available all data used to draw the conclusions outlined in your manuscript. Data must be deposited in an appropriate repository, included within the body of the manuscript, or uploaded as supporting information. This includes all numerical values that were used to generate graphs, histograms etc. For an example in PLOS Biology see here: http://www.plosbiology.org/article/info%3Adoi%2F10.1371%2Fjournal.pbio.1001908#s5.
---

## [Decision Letter · Decision Letter 1]

27 Feb 2023

Dear Dr. Besserve,

We are pleased to inform you that your manuscript 'Uncovering the Organization of Neural Circuits with Generalized Phase Locking Analysis' has been provisionally accepted for publication in PLOS Computational Biology.

Before or while celebrating, read this message until the end since it contains more nice comments and clarifications from one of the reviewers, as well as a spotted typo that you can fix in the proofing page.

Best regards,

Daniele Marinazzo

Section Editor

PLOS Computational Biology

Daniele Marinazzo

Section Editor

PLOS Computational Biology

Reviewer's Responses to Questions

**Comments to the Authors:**

Reviewer #1: I appreciate the authors’ engagement with my perhaps unreasonable demand for more careful framing, as well as the discussions that arose on what constitutes as mechanistic insight. Just to re-iterate, I think it’s a really cool piece of work that combines SFC analysis to mechanistic modeling to distill biophysical insight, instead of using it as a purely descriptive summary statistic as is the norm now, and carefully specifying the caveats and limitations of the analysis + model only make the results stronger and more convincing. I think the additional text clarifying the dependence between GPLA and a generative model in the intro and discussion are great and sufficient to address my concerns, and the additional analyses comparing to univariate PLA also provide examples where GPLA are demonstrably better in terms of interpretability. I have some small notes below, but the paper is good to go from my side.

- on response to RP1.4: sorry about the confusion, I just meant that another (simpler) method, such as univariate SFC, may also produce outputs that can be mapped consistently to values of the generative model’s parameters, and not anything deeper with regards to having to make high-level assumptions in the specific context of oscillogenisis. Of course, I agree with the authors that restricting to core hypotheses in the Lakotos sense is sensible and a common practice, and I have no problems with this. The response to RP1.5 addresses this sufficiently in minimizing general claims of superiority, and especially the additional text on how univariate measures can also take advantage of generative models.

- I’m happy to see the comparison against PLA in the included Figures 1-3, which more convincingly demonstrate a tangible difference between GPLA and existing methods, and agree that further SWR-specific analyses would make an interesting follow-up study.

- re: RP1.8, apologies for my confusion! Indeed, the inhibitory population affects the excitatory dynamics even if it’s not included in the LFP generation. I think it might be helpful to have subtitles for the subpanels of Fig5. (e.g., B: model of recurrent dynamics; C: model of e-field read-out…, etc.), as they all look quite similar to an inattentive reader (again, my apologies).

- the new results in response to RP1.10 looks promising, and yes, with the appropriate re-writing, I agree that it is not necessary to show that no other method can pick up differences, just that it is acknowledged that GPLA is not inherently special is its ability to do so given the generative model.

- In Figure 1B, the complex domain plot at the bottom (with the red and blue dots) appear to be missing in the in-line figure, but is fine in the separately-attached big version. Just a note for the authors to check this during proofing.

- typo for “guaranties” -> guarantees

Richard Gao

**Have the authors made all data and (if applicable) computational code underlying the findings in their manuscript fully available?**

Reviewer #1: Yes

PLOS authors have the option to publish the peer review history of their article (what does this mean?). If published, this will include your full peer review and any attached files.

Reviewer #1: **Yes: **Richard Gao

---

## [Editor Report · Acceptance letter]

28 Mar 2023

PCOMPBIOL-D-22-00931R1 

Uncovering the Organization of Neural Circuits with Generalized Phase Locking Analysis

Dear Dr Besserve,

I am pleased to inform you that your manuscript has been formally accepted for publication in PLOS Computational Biology. Your manuscript is now with our production department and you will be notified of the publication date in due course.

With kind regards,

Timea Kemeri-Szekernyes
